# Super-resolution microscopy compatible fluorescent probes reveal endogenous glucagon-like peptide-1 receptor distribution and dynamics

Julia Ast [1,2], Anastasia Arvaniti[1,2], Nicholas H.F. Fine [1,2], Daniela Nasteska [1,2], Fiona B. Ashford[1,2], Zania Stamataki [3], Zsombor Koszegi[1,2], Andrea Bacon[4], Ben J. Jones[5], Maria A. Lucey[5], Shugo Sasaki [6], Daniel I. Brierley [7], Benoit Hastoy [8], Alejandra Tomas[9], Giuseppe D'Agostino [10], Frank Reimann [11], Francis C. Lynn [6], Christopher A. Reissaus[12], Amelia K. Linnemann [12], Elisa D'Este[13], Davide Calebiro [1,2], Stefan Trapp [7], Kai Johnsson [14], Tom Podewin [14]*, Johannes Broichhagen [14]* & David J. Hodson [1,2]*

The glucagon-like peptide-1 receptor (GLP1R) is a class B G protein-coupled receptor (GPCR) involved in metabolism. Presently, its visualization is limited to genetic manipulation, anti-body detection or the use of probes that stimulate receptor activation. Herein, we present **LUXendin645**, a far-red fluorescent GLP1R antagonistic peptide label. **LUXendin645** produces intense and specific membrane labeling throughout live and fixed tissue. GLP1R signaling can additionally be evoked when the receptor is allosterically modulated in the presence of **LUXendin645**. Using **LUXendin645** and **LUXendin651**, we describe islet, brain and hESC-derived β-like cell GLP1R expression patterns, reveal higher-order GLP1R organization including membrane nanodomains, and track single receptor subpopulations. We furthermore show that the **LUXendin** backbone can be optimized for intravital two-photon imaging by installing a red fluorophore. Thus, our super-resolution compatible labeling probes allow visualization of endogenous GLP1R, and provide insight into class B GPCR distribution and dynamics both in vitro and in vivo.

[1] Institute of Metabolism and Systems Research (IMSR), and Centre of Membrane Proteins and Receptors (COMPARE), University of Birmingham, Birmingham, UK. [2] Centre for Endocrinology, Diabetes and Metabolism, Birmingham Health Partners, Birmingham, UK. [3] Centre for Liver Research, College of Medical and Dental Sciences, Institute for Immunology and Immunotherapy, University of Birmingham, Birmingham, UK. [4] Genome Editing Facility, Technology Hub, University of Birmingham, Birmingham, UK. [5] Division of Diabetes, Endocrinology and Metabolism, Section of Investigative Medicine, Imperial College London, London, UK. [6] Diabetes Research Group, BC Children's Hospital Research Institute, Vancouver, BC, Canada; Department of Surgery, University of British Columbia, Vancouver, BC, Canada. [7] Centre for Cardiovascular and Metabolic Neuroscience, Department of Neuroscience, Physiology & Pharmacology, University College London, London, UK. [8] Oxford Centre for Diabetes, Endocrinology & Metabolism, University of Oxford, Oxford, UK. [9] Division of Diabetes, Endocrinology and Metabolism, Section of Cell Biology and Functional Genomics, Imperial College London, London, UK. [10] Faculty of Biology, Medicine and Health, University of Manchester, Oxford Road, Manchester, UK. [11] Wellcome Trust-MRC Institute of Metabolic Science, University of Cambridge, Cambridge, UK. [12] Department of Pediatrics, Indiana University School of Medicine, Indianapolis, IN, USA. [13] Optical Microscopy Facility, Max Planck Institute for Medical Research, Heidelberg, Germany. [14] Department of Chemical Biology, Max Planck Institute for Medical Research, Heidelberg, Germany. *email: tom.podewin@mpimf-heidelberg.mpg.de; johannes.broichhagen@mr.mpg.de; d.hodson@bham.ac.uk

The glucagon-like peptide-1 receptor (GLP1R) is a secretin family class B G protein-coupled receptor (GPCR) characterized by hormone regulation[1]. Due to its involvement in glucose homeostasis, the GLP1R has become a blockbuster target for the treatment of type 2 diabetes mellitus[2]. The endogenous ligand, glucagon-like peptide-1 (GLP-1) is released from enteroendocrine L-cells in the gut in response to food intake[3], from where it travels to the pancreas before binding to its cognate receptor expressed in β-cells. Following activation, the GLP1R engages a cascade of signaling pathways including $Ca^{2+}$, cAMP, ERK and β-arrestin, which ultimately converge on β-cell survival and the glucose-dependent amplification of insulin release[4,5]. GLP1R is also expressed in the brain[6,7], where it further contributes to metabolism via effects on food intake, energy expenditure, locomotion, and insulin sensitivity. Despite this, GLP1R localization remains a challenge and is impeding functional characterization of GLP-1 and drug action.

Chemical biology and recombinant genetics have made available a diverse range of methods for the visualization of biological entities. Thus, classical fluorescent protein-fusions[8], self-labeling suicide enzymes (SNAP-, CLIP-, and Halo-tag)[9–11], "click chemistry"[12,13] and fluorogenic probes[14–16] have provided insight into the localization and distribution of their respective targets in living cells. In particular, current approaches for visualizing the GLP1R have so far relied on monoclonal antibodies (mAbs) directed against GLP1R epitopes[17,18], SNAP-tags[19–21], or fluorescent analogues of Exendin4(1–39), Exendin4(9–39), and Liraglutide[22–25]. Moreover, mouse models exist in which Cre recombinase is driven by the *Glp1r* promoter, allowing labeling of GLP1R-expressing cells when crossed with reporter mice[6,7].

Such methods have a number of shortcomings. Antibodies possess variable specificity[17] and tissue penetration, and GLP1R epitopes might be hidden or preferentially affected by fixation in different cell types and tissues. Enzyme self-labels allow GLP1R to be visualized in living cells without affecting ligand binding, but require heterologous expression and have therefore not yet been able to address endogenous receptor. Moreover, fluorescent analogues of Exendin4(1–39) and Liraglutide activate and internalize the receptor, which could confound results in live cells, particularly when used as a tool to sort purified populations (i.e. β-cells)[26,27]. Antagonist-linked fluorophores circumvent this issue, but the majority lack thorough pharmacological validation, or possess near infrared tags which require sophisticated confocal imaging modalities. On the other hand, reporter mouse strategies possess high fidelity, but cannot account for lineage-tracing artefacts, post-translational processing, protein stability and trafficking of native receptor[28]. Lastly, none of the aforementioned approaches are amenable to super-resolution imaging of endogenous GLP1R.

Given the wider reported roles of GLP-1 signaling in the heart[29], liver[30], immune system[2], and brain[31], it is clear that new tools are urgently required to help identify GLP-1 target sites, with repercussions for drug treatment and its side effects. In the present study, we therefore set out to generate a specific probe for endogenous GLP1R detection in its native, surface-exposed state in live and fixed tissue, without receptor activation. Herein, we report **LUXendin645** and **LUXendin651**, Cy5- and silicon rhodamine (SiR)- conjugated far-red fluorescent antagonists with excellent specificity, live tissue penetration, and super-resolution capability. Using our tools, we provide an updated view of GLP1R expression patterns in pancreatic islets, brain, and hESC-derived β-like cells, show that endogenous GLP1Rs form nanodomains at the membrane, and reveal receptor subpopulations with distinct diffusion modes in their non-stimulated state. Lastly, installation of a tetramethylrhodamine (TMR) fluorophore allows in vivo multiphoton imaging. As such, the **LUXendins** provide the first

**Table 1 Spectral properties of GLP1R labeling probes.**

| | Dye | $\lambda_{Ex}$ (nm) | $\lambda_{Em}$ (nm) | $\varepsilon^a$ ($M^{-1}$ $cm^{-1}$) | $\Phi$ |
|---|---|---|---|---|---|
| **LUXendin555** | TMR | 555 | 579 | 84,000 | 0.31 |
| **LUXendin645** | Cy5 | 645 | 664 | 250,000 | 0.22 |
| **LUXendin651** | SiR | 651 | 669 | 100,000 | 0.43 |

Maximal excitation and emission wavelengths, and quantum yields were acquired using probes dissolved at 10 μM in PBS, pH 7.4 at 21 °C
$^a$For maleimide-conjugated fluorophores

nanoscopic characterization of a class B GPCR, with wider flexibility for detection and interrogation of GLP1R in the tissue setting both in vitro and in vivo.

## Results

**Design of LUXendin555, LUXendin645, and LUXendin651**. Ideally, a fluorescent probe to specifically visualize a biomolecule should have the following characteristics: straightforward synthesis and easy accessibility, high solubility, relatively small size, high specificity and affinity, and a fluorescent moiety that exhibits photostability, brightness and (far-)red fluorescence with an additional two-photon cross-section. Moreover, the probe should be devoid of biological effects when applied to live cells and show good or no cell permeability, depending on its target localization. While some of these points were addressed in the past, we set out to achieve this high bar by designing a highly specific fluorescent GLP1R antagonist using TMR, Cy5, and SiR fluorophores. As no small molecule antagonists for the GLP1R are known, we turned to Exendin4(9–39), a potent antagonistic scaffold amenable to modification (Fig. 1)[32]. We used solid-phase peptide synthesis (SPPS) to generate an S39C mutant[21], which provides a C-terminal thiol handle for late-stage installation of different fluorophores. As such, TMR-, Cy5- and SiR-conjugated versions were obtained by means of cysteine-maleimide chemistry, termed **LUXendin555**, **LUXendin645**, and **LUXendin651**, respectively, with spectral properties shown in Table 1 (characterization of and purity of compounds in Supplementary Figs. 1–11) (Fig. 1).

**LUXendin645 intensely labels GLP1R in cells and tissue**. GLP-1-induced cAMP production was similarly blocked by Exendin4 (9–39) and its S39C mutant (Fig. 2a). Installation of Cy5 to produce **LUXendin645** did not affect these antagonist properties (Fig. 2a) (Supplementary Fig. 12). No agonist or partial agonist activity was detected for Exendin4(9–39), S39C_Exendin4(9–39), or **LUXendin645** (Fig. 2a) (Supplementary Fig. 12). As expected, addition of the GLP1R-positive allosteric modulator (PAM) BETP[33] conferred weak agonist activity on **LUXendin645** ($EC_{50}$(cAMP) = 192 nM) (Fig. 2b).

As a first assessment of GLP1R-labeling efficiency, we probed YFP-AD293-SNAP_GLP1R cells with increasing concentrations of **LUXendin645**. Maximal **LUXendin645** labeling occurred at 250–500 nM (Fig. 2c), with no signal detected in control YFP-AD293 cells lacking GLP1R (Fig. 2d). We next examined whether **LUXendin645** would allow labeling of endogenous GLP1R in primary tissue. Following 60 min application of **LUXendin645**, isolated islets demonstrated intense and clean labeling, which was restricted to the membrane (Fig. 2e). To minimize background fluorescence, slightly lower concentrations of **LUXendin645** were used in islets (50–100 nM) vs. plated cells (250 nM). Using conventional confocal microscopy, we were able to detect bright staining even 60 μm into the islet (Fig. 2e). Given these results, we next attempted to penetrate deeper into the islet by taking advantage of the superior axial resolution of two-photon

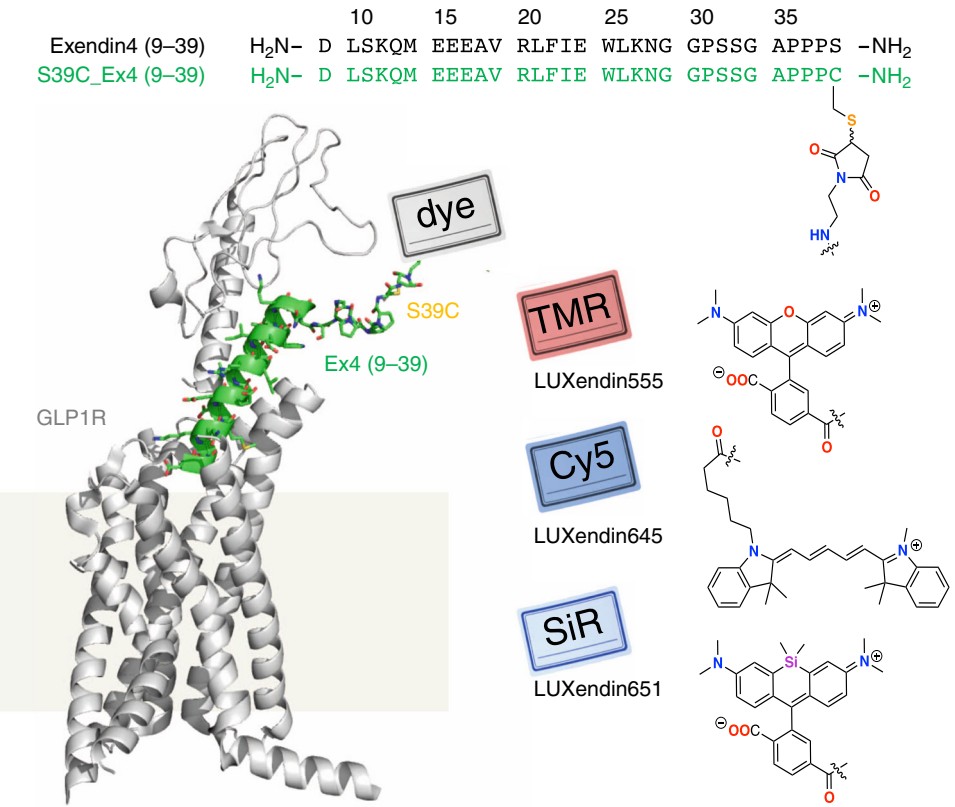

**Fig. 1 Sequence and structure of LUXendin555, LUXendin645, and LUXendin651.** LUXendins are based on the antagonist Exendin4(9–39), shown in complex with GLP1R. The label can be any dye, such as TMR (top), SiR (middle), or Cy5 (bottom) to give **LUXendin555, LUXendin645, and LUXendin651**, respectively. The model was obtained by using the cryo-EM structure of the activated form of GLP1R in complex with a G protein (pdb: 5VAI)[62], with the G protein and the 8 N-terminal amino acids of the ligand removed from the structure while mutating S39C and adding the respective linker. Models were obtained as representative cartoons by the in-built building capability of PyMOL (Palo Alto, CA, USA) without energy optimization. Succinimide stereochemistry is unknown and neglected for clarity.

excitation (Fig. 2f). Remarkably, this imaging modality revealed **LUXendin645** labeling at high resolution throughout the entire volume of the islet (170 μm in this case) (Fig. 2f). Consistent with the cAMP assays, GLP1R internalization was detected following co-application of **LUXendin645** and BETP to MIN6 β-cells, which endogenously express the receptor (Fig. 2g, h).

**LUXendin645 allows multiplexed GLP1R detection.** Demonstrating flexibility, **LUXendin645** labeling was still present following formaldehyde fixation (Fig. 2i, j). Immunostaining using a specific primary monoclonal antibody against the GLP1R revealed strong co-localization with **LUXendin645** in both islets (Fig. 2i) and MIN6 cells (Fig. 2j). Notably, **LUXendin645** displayed superior signal-to-noise-ratio and membrane resolution compared to the antibody (Fig. 2k), expected to be even better in live tissue where auto-fluorescence is less. Likewise, **LUXendin645** co-localized with SNAP-Surface 488 in SNAP_GLP1R-INS1 rat β-cells generated on an endogenous null background (Fig. 2l). Suggesting that **LUXendin645** requires the presence of surface GLP1R, labeling was markedly reduced following prior internalization with Exendin4(1–39) (94.1 ± 2.0% decrease in surface **LUXendin645** labelling with Exendin4(1–39) treatment, mean ± s.d.; $n = 3$ independent repeats) (Fig. 2l, m).

**LUXendin645 specifically binds the GLP1R.** To further validate the specificity of **LUXendin645** labeling in primary tissue, we generated *Glp1r* knock-out mice. This was achieved using CRISPR-Cas9 genome editing to introduce a deletion into exon 1

of the *Glp1r*. The consequent frameshift was associated with absence of translation and therefore a global GLP1R knockout, termed *Glp1r*[(GE)−/−], in which all intronic regions, and thus regulatory elements, are preserved (Fig. 3a, b). Wild-type (*Glp1r*[+/+]), heterozygous and homozygous littermates were phenotypically normal and possessed similar body weights (Fig. 3c).

Confirming successful GLP1R knock-out, insulin secretion assays in islets isolated from *Glp1r*[(GE)−/−] mice showed intact responses to glucose, but absence of Exendin4(1–39)-stimulated insulin secretion (Fig. 3d). Reflecting this finding, the incretin-mimetic Liraglutide was only able to stimulate cAMP rises in islets from wild-type (*Glp1r*[+/+]) littermates, measured using the FRET probe Epac2-camps (Fig. 3e, f). As expected, immunostaining with monoclonal antibody showed complete absence of GLP1R protein (Fig. 3g). Suggesting that **LUXendin645** specifically targets GLP1R, with little to no cross-talk from glucagon receptors[34], signal could not be detected in *Glp1r*[(GE)−/−] islets (Fig. 3g).

Together, these data provide strong evidence for a specific mode of **LUXendin645** action via the GLP1R.

**LUXendin645 highlights weak GLP1R expression.** Previous approaches have shown low abundance of *Glp1r* transcripts in the other major islet endocrine cell type, i.e. glucagon-secreting α-cells[7,35]. This is associated with detection of GLP1R protein in ~1–10% of cells[7,36], providing an excellent testbed for **LUXendin645** sensitivity and specificity. Studies in intact islets showed that **LUXendin645** labeling was widespread in the islet

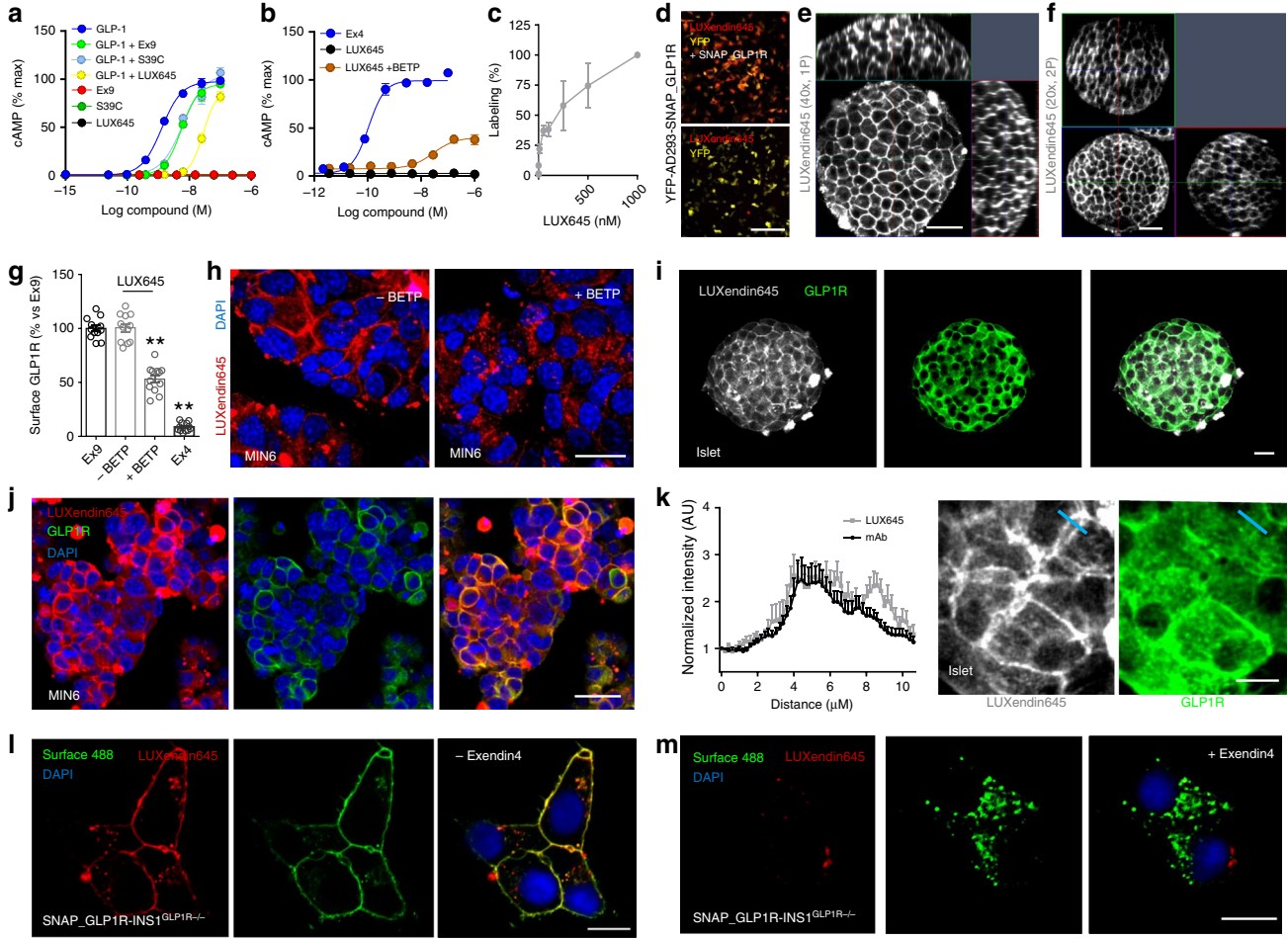

**Fig. 2 LUXendin645 binding, signaling, and labeling. a** Exendin4(9–39), S39C-Exendin4(9–39), and **LUXendin645** (LUX645) display similar antagonistic properties (applied at 1 µM) in HEK293-SNAP_GLP1R following 30 min GLP-1-stimulation ($n = 4$ independent assays). **b LUXendin645** weakly activates GLP1R in the presence of the positive allosteric modulator (PAM) BETP (25 µM) (30 min stimulation in HEK293-SNAP_GLP1R) (Ex4, +ve control) ($n = 4$ independent assays). **c LUXendin645** labels AD293-SNAP_GLP1R cells with maximal labeling at 250–500 nM ($n = 4$ independent assays). **d LUXendin645** signal cannot be detected in YFP-AD293 cells (scale bar = 212.5 µm) ($n = 3$ independent assays). **e** Representative confocal z-stack showing **LUXendin645** staining in a live islet ($n = 27$ islets, six animals, three separate islet preparations) (scale bar = 37.5 µm). **f** As for **e**, but two-photon z-stack (scale bar = 37.5 µm) (representative image from $n = 27$ islets, seven animals, three separate islet preparations). **g, h** 250 nM **LUXendin645** internalizes GLP1R in MIN6 β-cells when agonist activity is conferred using 25 µM BETP (Ex4 and Ex9 were applied at 100 and 250 nM, respectively) (scale bar = 21 µm) (representative images from $n = 12$ coverslips, three independent repeats) (one-way ANOVA with Bonferroni's test; $F = 217.6$, DF = 3). **i, j LUXendin645** signal co-localizes with a GLP1R monoclonal antibody in islets ($n = 13$ islets, three separate islet preparations) and MIN6 β-cells (representative images from $n = 24$ coverslips, three independent repeats) (scale bar = 26 µm). **k LUXendin645** improves membrane visualization compared to antibody (scale bar = 12.5 µm). Representative images are shown, with location of intensity-over-distance measures indicated in blue ($n = 18$ islets, five animals, three separate islet preparations). **l, m LUXendin645** co-localizes with Surface 488, pre-applied to Glp1r null SNAP_hGLP1R-INS1[GLP1−/−] cells **l**. Pre-treatment with Exendin4(1–39) to internalize the GLP1R reduces **LUXendin645**-labeling **m** (scale bar = 10 µm) (representative images from $n = 3$ independent repeats). **LUXendin645** was applied to cells at 250 nM and tissue at 50–100 nM. GLP-1 glucagon-like peptide-1; Ex9 Exendin4(9–39); S39C S39C-Exendin4(9–39); Ex4 Exendin4(1–39). Mean ± s.e.m. are shown. **$P < 0.01$ for all statistical tests. Source data are provided as a Source Data file.

and well co-localized with insulin immunostaining (Fig. 4a). However, **LUXendin645** could also be seen on membranes very closely associated with α-cells and somatostatin-secreting δ-cells (Fig. 4b, c), similarly to results obtained with GLP1R mAb. Due to the close apposition of β-, δ- and α–cell membranes, we were unable to accurately assign cell-type specificity to **LUXendin645**. Instead, using cell clusters plated onto coverslips, we could better discern **LUXendin645** labeling, revealing GLP1R expression in 12.3 ± 3.3% of α-cells (mean ± s.d.; $n = 18$ cell clusters, ten animals, three separate islet preparations) (Fig. 4d–f). Notably, GLP1R-expressing α-cells tended to adjoin, whereas those without the receptor were next to β-cells. Confirming previous findings, a majority (85.1 ± 16.3%) (mean ± s.d.; $n = 18$ cell clusters,

ten animals, three separate islet preparations) of β-cells were positive for **LUXendin645** (Fig. 4d–f)[7,25].

We wondered whether fixation required for immunostaining might increase background fluorescence such that GLP1R detection specificity was reduced. To circumvent this, studies were repeated in live islets where **LUXendin645** signal was found to be much brighter and background almost non-existent. GLP1R was detected in 24.6 ± 5.0% (mean ± s.d.; $n = 31$ islets, six animals, three separate islet preparations) of non-β-cells (Fig. 4g, h) using Ins1Cre[Thor];R26[mTmG] reporter mice in which β-cells are labeled green and all other cell types are labeled red following Cre-mediated recombination. Once adjusted for the previously reported GLP1R expression in δ-cells (assuming 100%), which

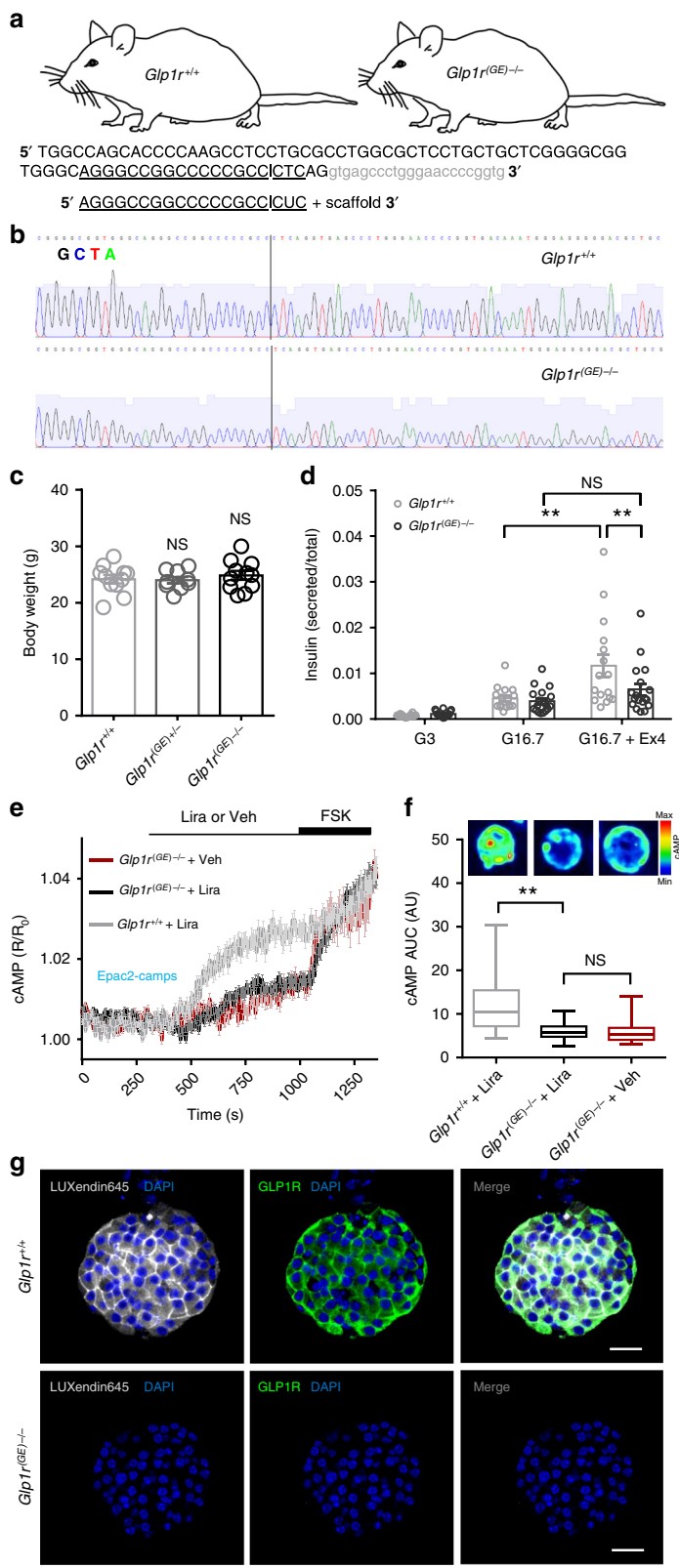

constitute ~20% of the insulin-negative islet population[37], this leaves ~5% of GLP1R+ α-cells. This was not an artefact of optical section, since two-photon islet reconstructions showed similar complete absence of **LUXendin645** staining in discrete regions near the surface (where α-cells predominate) (Supplementary Movie 1).

**LUXendins reveal higher-order GLP1R organization**. By combining **LUXendin645** with super-resolution radial fluctuations (SRRF) analysis[38], GLP1R could be imaged at super-resolution using streamed images (~500) from a conventional widefield microscope (Fig. 5a). To image endogenous GLP1R at <100 nm lateral resolution, we combined STED nanoscopy with

**Fig. 3 LUXendin645 is highly specific for the GLP1R. a** Schematic showing sgRNA-targeting strategy for the production of $Glp1r^{(GE)-/-}$ mice. The sgRNA used targeted $Glp1r$ and the double-strand break mediated by Cas9 lies within exon1 (capital letters); intron shown in gray. **b** $Glp1r^{(GE)-/-}$ animals harbor a single-nucleotide deletion, as shown by sequencing traces. **c** Body weights were similar in male 8–9 weeks old $Glp1r^{+/+}$, $Glp1r^{(GE)+/-}$, and $Glp1r^{(GE)-/-}$ littermates ($n = 9$ animals) (one-way ANOVA with Bonferroni's test; $F = 0.362$, DF = 2). **d** The incretin-mimetic Exendin4(1–39) (Ex4; 10 nM) is unable to significantly potentiate glucose-stimulated insulin secretion in $Glp1r^{(GE)-/-}$ islets ($n = 15$ repeats, six animals for each genotype, three separate islet preparations) (between genotype comparisons: two-way ANOVA with Sidak's test; $F = 4.061$, DF = 2) (within genotype comparisons: one-way ANOVA with Bonferroni's post-hoc test; $F = 14.57$ ($Glp1r^{+/+}$), 10.83 ($Glp1r^{(GE)-/-}$); DF = 2). **e** Liraglutide (Lira) does not stimulate cAMP beyond vehicle (Veh) control in $Glp1r^{(GE)-/-}$ islets, measured using the FRET probe Epac2-camps ($n = 25$ islets for each genotype, three animals per genotype, two separate islet preparations). **f** cAMP area-under-the-curve (AUC) quantification showing absence of significant Liraglutide-stimulation in $Glp1r^{(GE)-/-}$ islets ($n = 25$ islets for each genotype, three animals per genotype, two separate islet preparations) (Kruskal–Wallis test with Dunn's test; Kruskal–Wallis statistic = 31.78, DF = 2) (Box and Whiskers plot shows range and median) (representative images displayed above each bar; color scale shows min to max values as a ramp from blue to red). **g** LUXendin645 and GLP1R antibody labeling is not detectable in $Glp1r^{(GE)-/-}$ islets (scale bar = 40 μm) ($n = 27$ islets, five animals per genotype, three separate islet preparations). For all statistical tests, *$P < 0.05$, **$P < 0.01$ and NS, non-significant. In all cases, **LUXendin645** was applied at 100 nM. Mean ± s.e.m. are shown. Source data are provided as a Source Data file.

**LUXendin651**, which bears SiR instead of Cy5. **LUXendin651** displayed antagonist behavior, with no evidence of partial agonism, and produced bright labeling of wild-type but not $Glp1r^{(GE)-/-}$ islets, with an identical distribution to **LUXendin645** (Supplementary Figs. 12 and 13). Incubation of MIN6 cells with **LUXendin651** and subsequent fixation allowed STED imaging of the endogenous GLP1R with a FWHM = 70 ± 10 nm (mean ± s.d.; $n = 15$ line profiles measured on the raw data, from two independent repeats) (Fig. 5b–d). STED snapshots of MIN6 β-cells revealed detailed GLP1R distribution: receptors were not randomly arranged but rather tended to organize into nanodomains with neighbors (Fig. 5b–d). This was confirmed using the F- and G-functions, which showed a non-random and more clustered GLP1R distribution (Fig. 5e, f). Differences in GLP1R expression level and pattern could clearly be seen between neighboring cells with a subpopulation possessing highly concentrated GLP1R clusters (Fig. 5g). **LUXendin651** even allowed GLP1R to be imaged in living MIN6 cells using SRRF and STED, although nanodomains were more difficult to resolve due to the lateral diffusion of receptors (Fig. 5h, i).

**LUXendin645 and Luxendin651 label single GLP1R molecules.** To test whether **LUXendin645** and **LUXendin651** would be capable of tracking single GLP1Rs in live cells, we performed single-molecule microscopy experiments in which individual receptors labeled with either fluorescent probe were imaged by total internal reflection fluorescence (TIRF) microscopy[39,40]. Both probes allowed GLP1R to be tracked at the single-molecule level in CHO-K1-SNAP_GLP1R cells, but bleaching precluded longer recordings with **LUXendin645** (Fig. 6a and Supplementary Movies 2, 3). By combining single-particle tracking with **LUXendin651**, we were able to show that GLP1Rs diffuse at the membrane in their non-stimulated or antagonized state (Fig. 6a and Supplementary Movie 4). However, a mean square displacement (MSD) analysis[40] revealed a high heterogeneity in the diffusion of GLP1Rs on the plasma membrane, ranging from virtually immobile receptors to some displaying features of directed motion (superdiffusion) (Fig. 6b, c). GLP1R diffusion properties were not ligand-dependent, since similar profiles were detected for both **LUXendin645** and **LUXendin651** (Fig. 6c).

**LUXendin645 allows visualization of central GLP1 targets.** To further show the utility of **LUXendin645** for visualizing endogenous GLP1R, we extended studies to the brain in which mAbs do not work reliably and where peripheral GLP1 targets still remain poorly characterized. Two hours following subcutaneous injection of **LUXendin645**, perfuse-fixed brains were retrieved for analysis. Intense labelling could be detected in the arcuate nucleus

(ARC), area postrema (AP) and choroid plexus (CP) (Fig. 7a, b), all regions known to express GLP1R using reporter or fluorescent agonist approaches[6,22]. Notably, **LUXendin645**-labeled neurons overlapped with areas receiving innervation from GLP1-producing neurons[41], with GLU-YFP synaptic boutons closely abutting GLP1R+ areas (Fig. 7a, b). **LUXendin645** labeling co-localized with GLP1R-expressing neurons in the ARC/median eminence (ME) and AP/nucleus tractus solitarius (NTS), shown using GLP1RCre;LSL-GCaMP3 reporter mice (Fig. 7c–e).

Super-resolution imaging (~140 nm lateral resolution) revealed the presence of **LUXendin645** on the cell membranes of GLP1R+ neuron cell bodies, as well as dendrites. Moreover, GLP1R were found to accumulate into nanodomains on the membranes of ARC and AP neuron membranes, as well as ependymal cells of the CP (Fig. 7f). Lastly, optical projection tomography allowed entire **LUXendin645**-labelled brains to be imaged and mapped in three-dimensions, confirming the above results and also extending probe localization to the subfornical organ (SFO), organum vasculosum of the lamina terminalis (OVLT), and ventricles (Fig. 7g).

**LUXendin645 labels GLP1R in hESC-derived β-cells.** Since GLP1R are expressed in mature β-cells[42], we wondered whether **LUXendin645** would serve as a useful surface marker for assessing differentiation of human embryonic stem cell (hESC)-derived β-like cells. **LUXendin645** was unable to label undifferentiated ES cells (Fig. 8a). Following differentiation and 21 days' culture, **LUXendin645** labeling was clearly visible in spheroids, in line with increasing levels of $GLP1R$ expression (Fig. 8b). As for mouse islets, **LUXendin645** co-localized with insulin, with minimal signal in areas strongly positive for glucagon (Fig. 8c), shown using Manders' split coefficients (Fig. 8d). Not all insulin-containing cells stained for GLP1R, however (Fig. 8c). Confirming a β-like cell phenotype, spheroids were fixed and sliced before staining for insulin and NKX6-1 (Fig. 8e).

We next investigated whether **LUXendin645** would allow β-like cells to be purified according to GLP1R expression. To this end, **LUXendin645**-labelled spheroids were subjected to fluorescence-activated cell sorting (FACS), before gene expression analyses of **LUXendin645**+ and **LUXendin645**− populations. Notably, the **LUXendin645**+ population expressed higher levels of $GLP1R$ and $NKX6-1$, with a tendency toward increased $INS$ (Fig. 8f). As expected from the imaging data, $GCG$ expression was significantly decreased in the **LUXendin645**+ β-like cells (Fig. 8f).

**LUXendin555 allows labeling of islets in vivo.** Lastly, we explored whether swapping the far-red Cy5/SiR for a TMR dye would be tolerated to obtain a spectrally orthogonal probe,

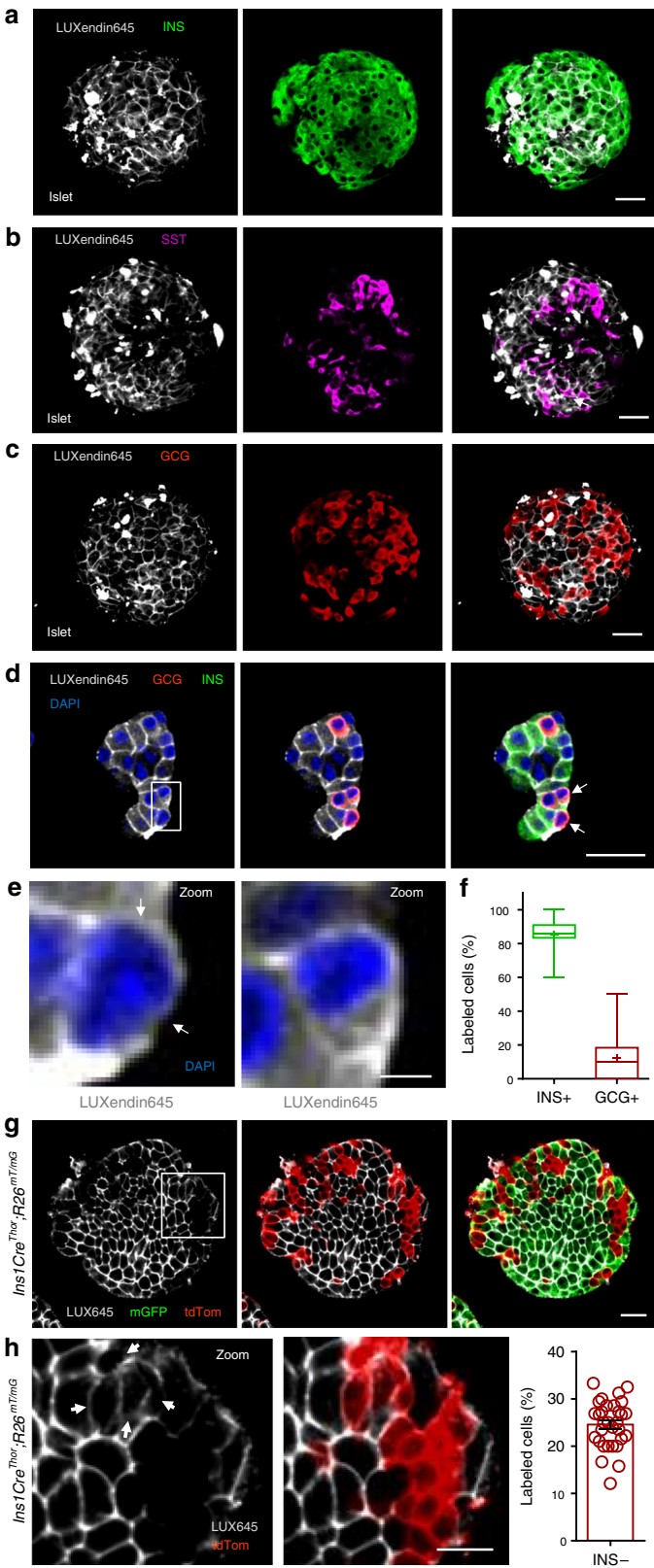

termed **LUXendin555**. Labeling was detected in YFP-AD293-SNAP_GLP1R (Fig. 9a) but not in YFP-AD293 cells (Fig. 9b), with max labeling at 600 nM (Fig. 9c). A slightly lower concentration (250 nM) of **LUXendin555** was found to produce bright staining in both cells and islets, whilst minimizing background fluorescence. However, we noticed a more punctate

**LUXendin555** staining pattern when viewed at high-resolutions (Fig. 9a). To determine whether the appearance of puncta was due to receptor internalization, or alternatively accumulation of cleaved, charged TMR in organelles, labeling was repeated in islets co-stained with GLP1R monoclonal antibody (Supplementary Fig. 14). In these experiments, no differences in GLP1R

**Fig. 4 LUXendin645 reveals GLP1R expression in a subpopulation of α-cells. a–c** LUXendin645 labeling is widespread throughout the intact islet, co-localizing predominantly with β-cells **a** and δ-cells **b**, but less so with α-cells **c** stained for insulin (INS), somatostatin (SST), and glucagon (GCG), respectively ($n = 18$ islets, seven animals, three separate islet preparations) (scale bar $= 26\,\mu m$). **d** Following dissociation of islets into cell clusters, **LUXendin645** labeling can be more accurately quantified (arrows highlight cells selected for zoom-in) (scale bar $= 26\,\mu m$). **e** Zoom-in of **d** showing a **LUXendin645**− (left) and **LUXendin645**+ (right) α-cell (arrows highlight non-labeled cell membrane, which is not bounded by a β-cell) (scale bar $= 26\,\mu m$). **f** Box-and-whiskers plot showing proportion of β-cells (INS) and α-cells (GCG) co-localized with **LUXendin645** ($n = 18$ cell clusters, ten animals, three separate islet preparations) (box and whiskers plot shows range and median; mean is shown by a plus symbol). **g** $Ins1Cre^{Thor};R26^{mT/mG}$ dual fluorophore reporter islets express tdTomato until Cre-mediated replacement with mGFP, allowing identification of β-cells (~80% of the islet population) and non-β-cells for live imaging (scale bar $= 26\,\mu m$). **LUXendin645** (LUX645) highlights GLP1R expression in nearly all β-cells but relatively few non-β-cells ($n = 31$ islets, six animals, three separate islet preparations). **h** A zoom-in of the islet in **g** showing GLP1R expression in some non-β-cells (left) together with quantification (right) (arrows show **LUXendin645**-labeled non-β cells) (scale bar $= 12.5\,\mu m$) (scatter dot plot shows mean ± s.e.m.). White boxes show the location of zoom-ins. In all cases, **LUXendin645** was applied at 100 nM. Source data are provided as a Source Data file.

surface expression could be seen between **LUXendin555**, **LUXendin645**, and Ex9 (Fig. 9d), suggesting that puncta are unlikely to be internalized receptor. In line with this, **LUXendin555** was found to display antagonist properties using HTRF-based cAMP assay (Fig. 9e). However, a second independent detection method (luciferase) showed the opposite result, raising the possibility that the installed TMR might influence assay readout (Fig. 9f). As for the other probes, **LUXendin555** was unable to label $Glp1r^{(GE)-/-}$ islets (Supplementary Fig. 14).

We thought that the relatively high quantum yield of TMR, coupled with good two-photon cross-section might suit **LUXendin555** well to in vivo imaging. Two-photon imaging was applied to anaesthetized mice to allow visualization of the intact pancreas, exposed through an abdominal incision (Fig. 9g). Vessels and nuclei were first labeled using FITC-dextran and Hoechst before injecting **LUXendin555** intravenously. The following observations support that **LUXendin555** displays antagonist activity in vivo: (1) labeling occurred rapidly within 5 min post-injection; (2) staining was confined to the cell membrane with no apparent internalization (Fig. 9h); and (3) normoglycemia was not significantly altered over 30 min (173.0 ± 21.1 vs. 215.3 ± 41.4 mg/dl, 0 and 30 min post-injection, respectively; mean ± s.d.; $n = 3$ mice; non-significant, paired $t$-test).

## Discussion

In the present study, we synthesize and validate far-red fluorescent labels, termed **LUXendins** for the real-time detection of endogenous GLP1R in live cells. Nanomolar concentrations of **LUXendin645** and **LUXendin651** lead to intense membrane-labeling of the GLP1R, with best in class tissue penetration and signal-to-noise ratio, as well as super-resolution capability. Notably, **LUXendin645** and **LUXendin651** do not activate the GLP1R unless agonist activity is conferred with the widely available PAM BETP. **LUXendin645** and **LUXendin651** are highly specific, as shown using a CRISPR-Cas9 mouse line lacking GLP1R expression. Lastly, the analogous compound **LUXendin555**, bearing a different fluorophore, expands the color palette without changing the peptidic pharmacophore.

Compared to present chemical biology approaches, **LUXendins** possess a number of advantages for GLP1R labeling, which generally rely on Exendin4(1–39) or Exendin4(9–39) peptides labeled with for instance FITC, Cy3, Alexa594, Cy5, or VT750[19,22–25,33]. Firstly, the use of an antagonist retains more receptor at the cell surface, which likely increases detection capability. Secondly, the GLP1R is not fully activated, meaning that results can be interpreted in the absence of potentially confounding cell signaling or internalization, such as that expected with agonists[25]. Thirdly, Cy5 and SiR occupy the far-red range, leading to less background fluorescence, increasing depth penetration in confocal microscopy due to reduced scatter, and avoiding the use of more phototoxic wavelengths[43]. Fourthly,

**LUXendin** pharmacology and labelling specificity has been validated in-depth. Lastly, **LUXendins** are well-adapted for super-resolution imaging through the use of optimally suited fluorescent moieties. Together, these desirable properties open up the possibility to image expression and distribution of native GLP1R over extended periods of time using multiple imaging modalities.

While **LUXendins** also allowed GLP1R trafficking to be monitored, this required the presence of a PAM to allosterically activate the receptor. Due to the probe-dependent nature of PAMs, **LUXendins** with a number of different pharmacophores would need to be generated to fully assess the ligand-dependency of GLP1R trafficking. In some experiments, we also noticed the presence of punctate **LUXendin645** and **LUXendin651** labelling. Suggesting that this staining pattern reflects cleaved fluorophore rather than internalized GLP1R are the following observations: (1) succinimide exchange with reactive thiols can lead to linker loss[44], allowing free fluorophore to cross the membrane and accumulate in organelles; and (2) puncta were not apparent in the same samples co-stained with GLP1R mAb.

To test the specificity of **LUXendins**, we used CRISPR-Cas9 genome-editing to globally knock out the GLP1R in mice. Protein deletion was confirmed by absence of detectable GLP1R signal following labeling with monoclonal antibody, **LUXendin555**, **LUXendin645**, and **LUXendin651**. While $Glp1r^{-/-}$ animals already exist, and have made important contributions to our understanding of incretin biology, they were produced using a mutation to replace exons encoding transmembrane regions 1 and 3 (encoded by exons 5 and 7), presumably leading to deletion of the introns in between (~6.25 kb)[45]. By contrast, $Glp1r^{(GE)-/-}$ mice possess intact introns. Since introns contain regulatory elements, such as distant-acting enhancers[46], miRNAs[47], and lncRNAs[48], their loss in transgenic knockouts could have wider influence on the transcriptome. GLP1R knock-out mice might therefore be useful alongside conventional approaches for validating GLP1R reagents, including antibodies, agonist and antagonist, and derivatives thereof.

Demonstrating the excellent sensitivity of the Cy5-linked **LUXendin645** in particular, we were able to detect GLP1R expression in ~5% of α-cells. Understanding α-cell GLP1R expression patterns is important because incretin-mimetics reduce glucagon secretion[49], which would otherwise act to aggravate blood glucose levels. Previous studies using antibodies, reporter animals and agonist-fluorophores have shown that ~1–10% of mouse and rat α-cells express GLP1R, in line with the low transcript abundance[7,25,35,50], despite reports that GLP-1 can directly suppress glucagon release[36]. Our data are in general concordance with these findings, but demonstrate an increase in sensitivity compared to other approaches capable of detecting native GLP1R protein, namely mAb and agonist-fluorophore. This improvement is likely related to the superior signal-to-noise ratio (SNR) and selectivity of **LUXendin645**, increasing the

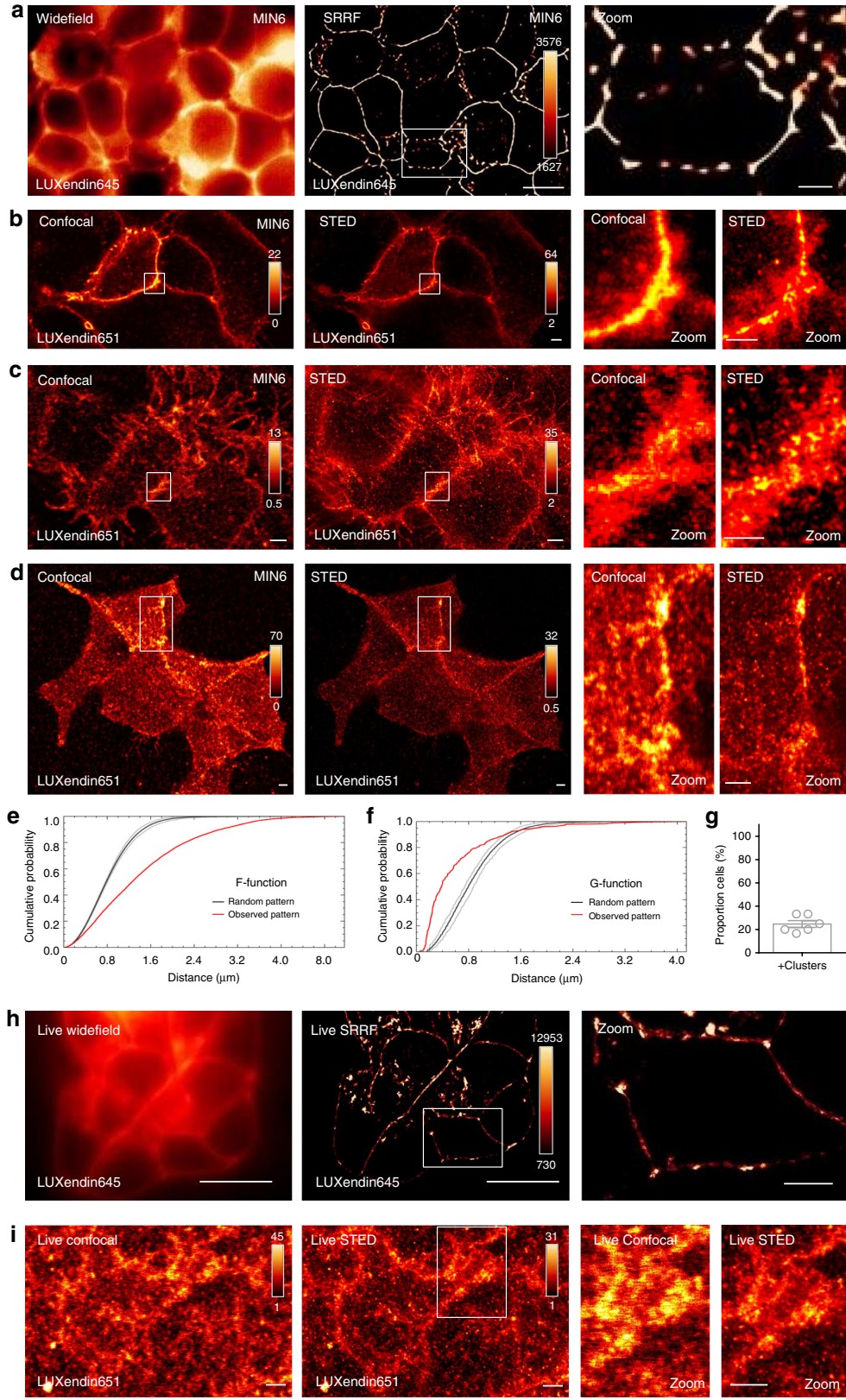

ability to resolve relatively low levels of endogenous GLP1R. A recent report showed GLP1R expression in ~80% of α-cells using an antibody raised against the *N*-terminal region, with both membrane and cytosolic staining evident[51]. While the reasons for this discrepancy are unknown, it should be noted that

**LUXendin645** binds the orthosteric site and so reports the proportion of GLP1R that is "signaling competent"[7,19,32].

In addition to islets, **LUXendin645** was also able to label brain tissue. 3D rendering of entire cleared brains using optical projection tomography showed strong **LUXendin645** staining in the

**Fig. 5 LUXendin651 and LUXendin645 allow nanoscopic detection of GLP1R. a** LUXendin645 allows super-resolution snapshots of MIN6 β-cells using widefield microscopy combined with super-resolution radial fluctuations (SRRF) (representative image from $n = 8$ images, three independent repeatss) (scale bar = 10 μm for full-field images, 2.5 μm for zoomed-in images). **b–d** Confocal and STED snapshots of endogenous GLP1R in **LUXendin651**-treated MIN6 cells at FWHM = 70 ± 10 nm (mean ± s.d.; $n = 15$ line profiles measured on the raw data, two independent repeats). Note the presence of punctate GLP1R expression as well as aggregation/clustering in cells imaged just away from **b**, close to **c** or next to **d** the coverslip using STED microscopy (representative image from $n = 8$ images, three independent repeats) (scale bar = 2 μm for full-field images, 1 μm for zoomed-in images). **e, f** Representative graph showing spatial analysis of GLP1R expression patterns using the F-function **e** and G-function **f**, which show distribution (red line) vs. a random model (black line; 95% confidence interval shown) ($n = 6$ from three independent repeats). **g** Approximately 1 in 4 MIN6 β-cells possess highly concentrated GLP1R clusters. **h, i** LUXendin651 allows GLP1R to be imaged in living MIN6 cells using SRRF **h** and STED **i** (representative image from $n = 6$ and 18 images, three independent repeats for SRRF and STED, respectively) (scale bar = 10 μm for full-field SRRF image, 2.5 μm for the zoomed-in image) (scale bar = 2 μm for STED images). White boxes show the location of zoom-ins. The following compound concentrations were used: 100 nM **LUXendin645** (SRRF) and 100–400 nM **LUXendin651** (STED). Mean ± s.e.m. are shown. Source data are provided as a Source Data file.

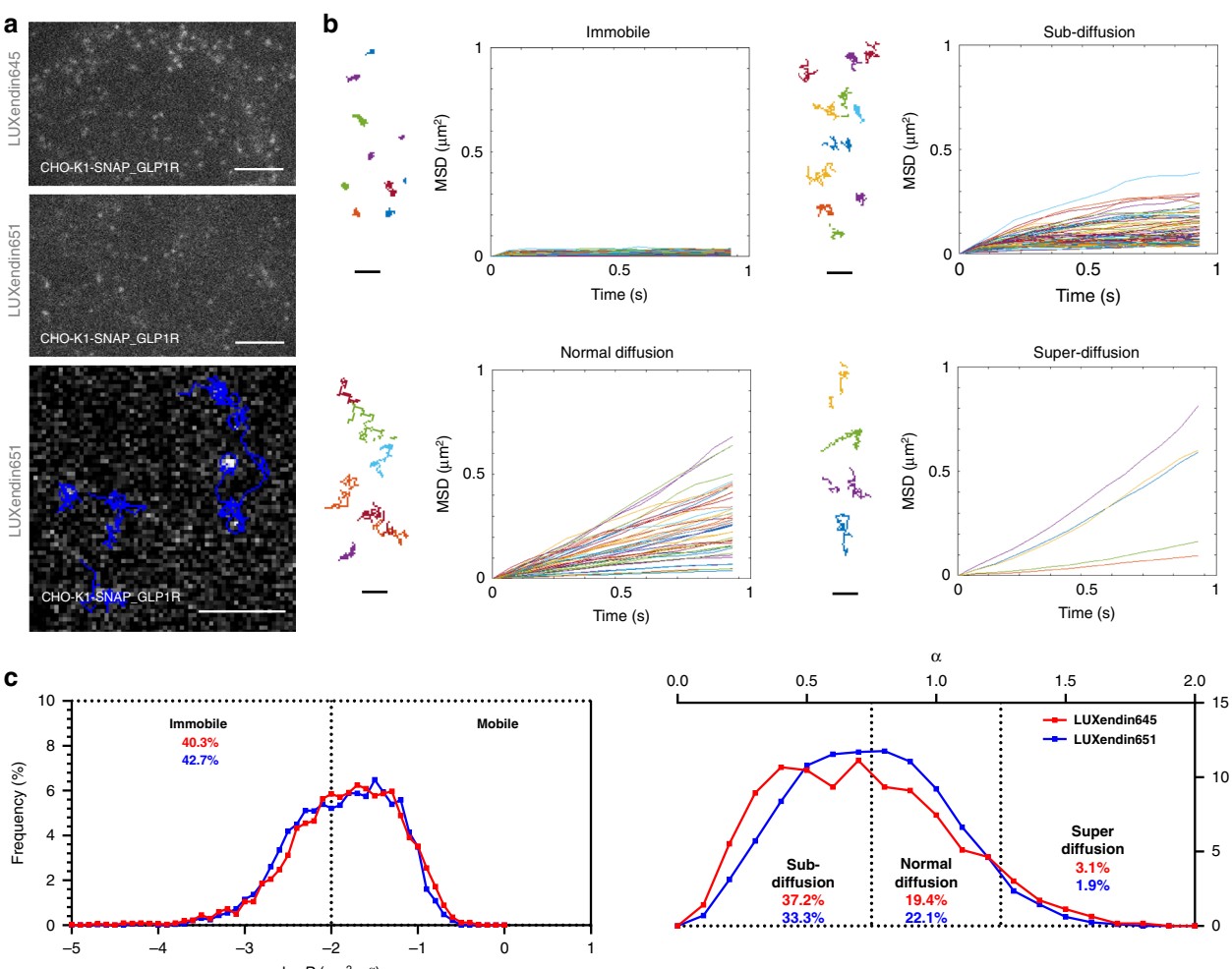

**Fig. 6 LUXendin645 and LUXendin651 allow single molecule GLP1R imaging. a** Representative single molecule microscopy images showing tracking of **LUXendin645**- and **LUXendin651**-labeled GLP1R at or close to the membrane (scale bar = 3 μm). **b** Mean square displacement (MSD) analysis showing different GLP1R diffusion modes (representative trajectories are displayed) (scale bar = 1 μm). **c** GLP1R molecules with diffusion coefficient $D < 0.01$ are classed as immobile (left), whilst those with $D > 0.01$ are further divided according to their anomalous diffusion exponent ($\alpha$), which defines the type of motion followed (confined, normal, or directed) (right) (pooled data from $n = 16$ cells, 5057–8612 trajectories, six independent repeats). **LUXendin645** and **LUXendin651** were used at 100 pM. Source data are provided as a Source Data file.

circumventricular organs and their directly neighboring regions, including the NTS and ARC. We also detected **LUXendin645**-labelling in the CP, an epithelial-vascular structure that secretes cerebrospinal fluid (CSF). Rat CP has recently been shown to express GLP1R ex vivo, with Ex4 reducing intracranial pressure in hydrocephalic models[52]. Notably, super-resolution snapshots showed that GLP1R in the ARC, AP, and CP were organized

as nanodomains at the membrane. Such higher organization has not been previously appreciated in the brain and it will now be interesting to understand the functional relevance for GLP1 signaling. A number of central GLP1R-expressing regions remained obscure, likely reflecting the pharmacokinetic properties of peripherally administered agonist/antagonist. However, we anticipate that **LUXendin645** will be useful for the study of these

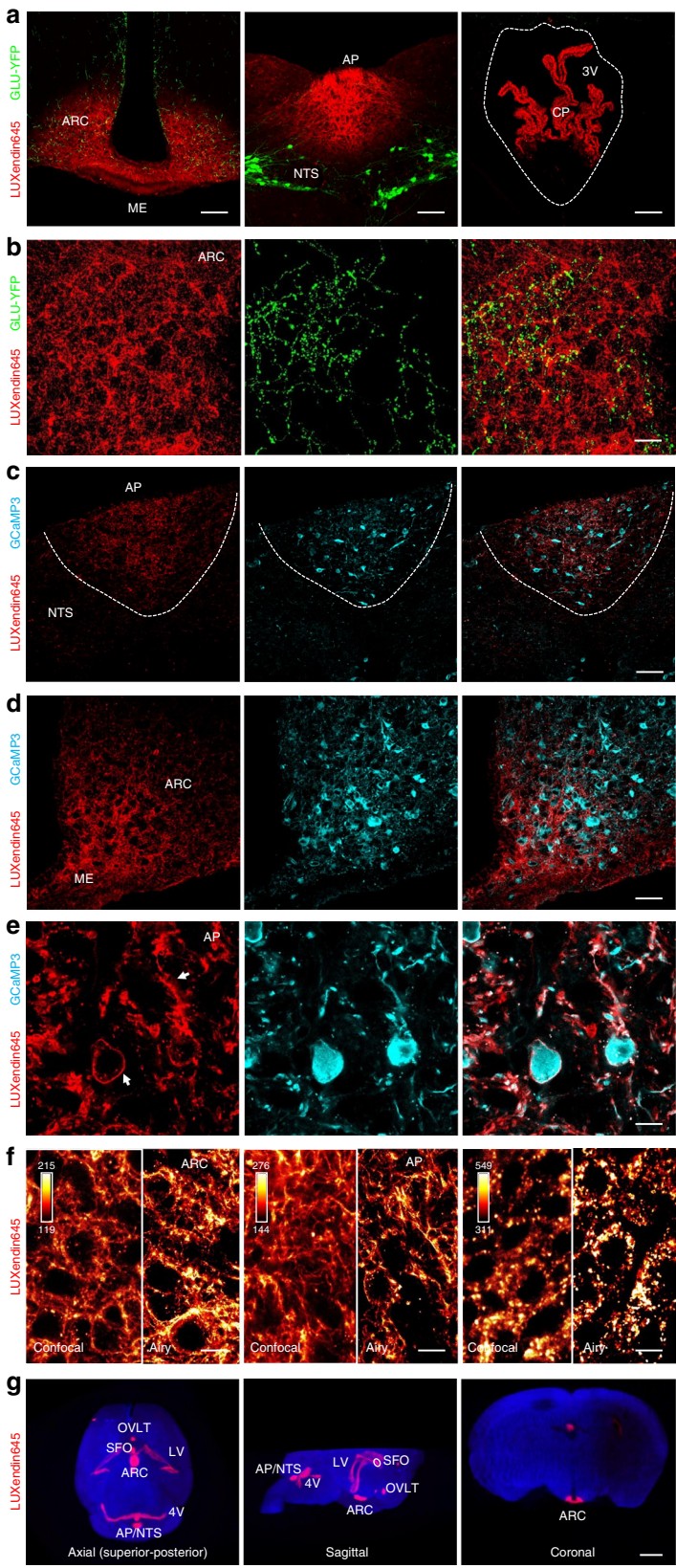

regions following targeted injections. Taken together, these data show that **LUXendin645** allows endogenous GLP1R to be visualized in extrapancreatic tissue using both conventional and super-resolution-imaging approaches.

We could also extend studies to hESC-derived β-like cells, where GLP1R expression was detected in spheroids following induction of differentiation and 21 days' culture. **LUXendin645** labeling was seen almost exclusively in the β-like cell compartment, reflecting known distributions in mouse[7,25] and human[53] islets. While not all β-like cells were visibly labelled, this likely reflects GLP1R expression levels, which were ~50% lower in spheroids than adult human islets. Pertinently, **LUXendin645**

**Fig. 7 LUXendin645 highlights GLP1R-expressing neurons in the brain. a** LUXendin645 labeling is detected in the the median eminence (ME), arcuate nucleus (ARC), area postrema (AP)/nucleus tractus solitaris (NTS), and choroid plexus (CP), in close association with GLP1-producing neurons, identified using *GLU-YFP* reporter animals (3V, third ventricle) (representative images from $n = 4$ animals) (scale bar = 106 μm). **b** Z-projection of an image stack (~30 μm) showing direct contacts between **LUXendin645**-labelled and GLP1-producing (GLU-YFP) neurons in the ARC (representative image from $n = 4$ animals) (scale bar = 20 μm). **c, d** LUXendin645 labeling co-localizes with GLP1R+ neurons in the AP/NTS **c** and ARC **d**, identified using *GLP1RCre;LSL-GCaMP3* reporter animals (representative image from $n = 4$ animals) (scale bar = 61 μm). **e** Super-resolution imaging using Airyscan shows that **LUXendin645** labeling is restricted to the membrane of the cell body and dendrites of GLP1R+ neurons (arrows show cell body and dendrite from left to right, respectively) (representative images from $n = 4$ animals) (scale bar = 9 μm). **f** GLP1R form nanodomains in ARC and AP neurons, as well as ependymal cells of the CP (confocal image is shown on the left for comparison) (representative images from $n = 8$ animals) (scale bar = 9 μm). **g** Mapping of **LUXendin645** distribution in cleared brains shows labelling of the ARC, AP/NTS, CP, lateral ventricles (LV), fourth ventricle (4V), subfornical organ (SFO), and organum vasculosum of the lamina terminalis (OVLT) (representative images from $n = 4$ animals) (scale bar = 1 mm). Note that, due to suspension of the brain, the coronal section is slightly offset in the dorsal–ventral plane; hence, the SFO appears above the ARC. In all cases, **LUXendin645** was injected subcutaneously at 100 pmol/g.

allowed more mature/differentiated β-like cells to be identified and purified according to GLP1R, which increases almost 20-fold in adult vs. neonatal rat β-cells[42]. Thus, these studies show the potential of **LUXendins** to understand GLP1R expression patterns in regenerated β-like cells, as well as rapidly mark differentiation/maturity status using a simple one-step surface marker.

Since **LUXendin645** showed excellent SNR using conventional epifluorescence, it was highly amenable to SRRF analysis. As such, **LUXendin645** and its congeners open up the possibility to image the GLP1R at super-resolution using simple widefield microscopy available in most laboratories. For stimulated emission depletion (STED) microscopy experiments, Cy5 was replaced with SiR to give **LUXendin651**. STED imaging showed that endogenous GLP1R possess a higher structural order in the presence of **LUXendin651** binding: namely organization into nanodomains at the cell membrane. Since GLP1R plasma membrane distribution is ligand-dependent, for example via effects on palmitoylation and clustering into cholesterol-rich nanodomains[20] it will be interesting to repeat these experiments using a range of agonists/antagonists. Indeed, **LUXendins** provide an ideal template for the production of fluorescent ligands that would allow super-resolution examination of nanodomain architecture in response to different activation modes.

Notably, a subpopulation of β-cells appeared to possess concentrated GLP1R clusters even in unstimulated conditions. It will be important in the future to investigate whether this is a cell autonomous heterogenous trait, or instead reflects biased distribution of receptors in membranes of specific β-cells. Lastly, both **LUXendin645** and **LUXendin651** allowed GLP1Rs to be imaged in live cells by single molecule microscopy, revealing variability in diffusion at the plasma membrane. Particle tracking analyses segregated GLP1R into four different populations based upon diffusion mode, in keeping with data from a class A GPCR, the beta adrenergic receptors[40]. Together, these experiments provide the first super-resolution characterization of a class B GPCR and suggest a degree of complexity not readily appreciated with previous approaches.

**LUXendin555** showed antagonist activity in terms of cAMP generation using a HTRF approach. However, we could not confirm this result using a luciferase-based detection method. The reasons for this are unclear, but might include interference of the red fluorescent TMR with either assay, or alternatively different GLP1R coupling strength between cell lines. As such, use of **LUXendin555** should consider the possibility that the ligand is an antagonist or agonist. Nonetheless, **LUXendin555** retained GLP1R at the membrane, and possesses advantageous properties for in vivo imaging including good two-photon cross-section and high quantum yield.

In summary, we provide a comprehensively tested and unique GLP1R detection toolbox consisting of far-red antagonist labels,

**LUXendin645** and **LUXendin651**, an agonist/antagonist **LUXendin555**, and knockout $Glp1r^{(GE)-/-}$ animals. Using these freely available probes, we provide an updated view of GLP1R organization, with relevance for the treatment of complex metabolic diseases such as obesity and diabetes. Thus, the stage is set for visualizing GLP1R in various tissues using a range of imaging techniques, as well as the production of peptidic labels and agonists.

## Methods

**Synthesis.** A free cysteine bioconjugation handle was installed on Exendin4(9–39) using solid-phase synthesis to give the derivatized S39C-Exendin4(9–39)[21]. Maleimide-conjugated-6-TMR, -6-SiR and -Cy5 were obtained by TSTU activation of the corresponding acids and reaction with 1-(2-amino-ethyl)-pyrrole-2,5-dione (TFA salt, Aldrich). Fluorophore coupling via thiol-maleimide chemistry to peptides was performed in PBS. All compounds were characterized by HRMS and purity was assessed to be >95% by HPLC. Extinction coefficients were based upon known manufacturer bulk material measures for TMR-Mal, Cy5-Mal (both Lumiprobe), and SiR-Mal (Spirochrome). Details for synthesis including characterization of all **LUXendins** are detailed in the Supplementary Methods and Supplementary Figs. 1–11. **LUXendin555**, **LUXendin645**, and **LUXendin651** are freely available for academic use upon request.

**Cell culture.** AD293 cells (Agilent) were maintained in Dulbecco's Modified Eagles medium (DMEM; D6546, Sigma) supplemented with 10% fetal calf serum (FCS), 1% L-glutamine, and 1% penicillin/streptomycin. CHO-K1 cells stably expressing the human SNAP_GLP1R (Cisbio) (CHO-K1-SNAP_GLP1R) were maintained in DMEM supplemented with 10% FCS, 1% penicillin/streptomycin, 500 μg/mL G418, 25 mM HEPES and 2% nonessential amino acids and 2% L-glutamine. MIN6 β-cells (a kind gift from Prof. Jun-ichi Miyazaki, Osaka University) were maintained in DMEM (D6546, Sigma) supplemented with 15% FCS, 25 mM D-glucose, 71 μM BME, 2 mM L-glutamine, 100 U/mL penicillin, 100 μg/mL streptomycin, and 25 mM HEPES. INS1 832/3 CRISPR-deleted for the endogenous GLP1R locus (a kind gift from Dr. Jacqui Naylor, MedImmune)[54] were transfected with human SNAP_GLP1R, before FACS of the SNAP-Surface488-positive population and selection using G41820. The resulting SNAP_GLP1R_INS1$^{GLP1R−/−}$ cells were maintained in RPMI-1640 supplemented with 10% FBS, 10 mM HEPES, 2 mM L-glutamine, 1 mM pyruvate, 72 μM β-mercaptoethanol, 1% penicillin/streptomycin, and 500 μg/mL G418.

**Animals.** $Glp1r^{(GE)-/-}$: CRISPR-Cas9 genome-editing was used to introduce a single base pair deletion into exon 1 of the $Glp1r$ locus. Fertilized eggs of female Cas9-overexpressing mice (strain $Gt(ROSA)26Sor^{tm1.1(CAG-cas9*,-EGFP)Fezh}$/J; JAX stock no. 024858) were harvested following super-ovulation. Modified single-guide RNA (Synthego) targeting exon 1 of $Glp1r$ and a single-stranded repair-template were injected at 20 ng/μl into the pronucleus of embryos at the 1-cell stage. In culture, 80% of embryos reached the 2-cell stage and were transplanted into surrogate mice. $Glp1r^{(GE)-/-}$ mice did not integrate the repair-template (confirmed by genotyping PCR), but instead harbored a single nucleotide deletion leading to a frame-shift mutation and loss of GLP1R protein. Knock-in mice that integrated the repair template were not used in the present studies and will be described elsewhere. The targeted locus of $Glp1r^{(GE)-/-}$ offspring was analyzed by PCR and sequencing. Off-target sites were predicted using the CRISPR Guide Design Tool (crispr.mit.edu). Loci of the top 10 off-target hits were amplified by PCR and analyzed via Sanger sequencing (Supplementary Table 1). Founder animals carrying alleles with small deletions were backcrossed to wild type animals (strain C57BL/6J) for 1–3 generations to outbreed affected off-targets and then bred to homozygosity (Supplementary Figs. 15 and 16). Animals were born in Mendelian ratios, genotyping was performed using Sanger sequencing or PCR. Genotyping PCRs were performed

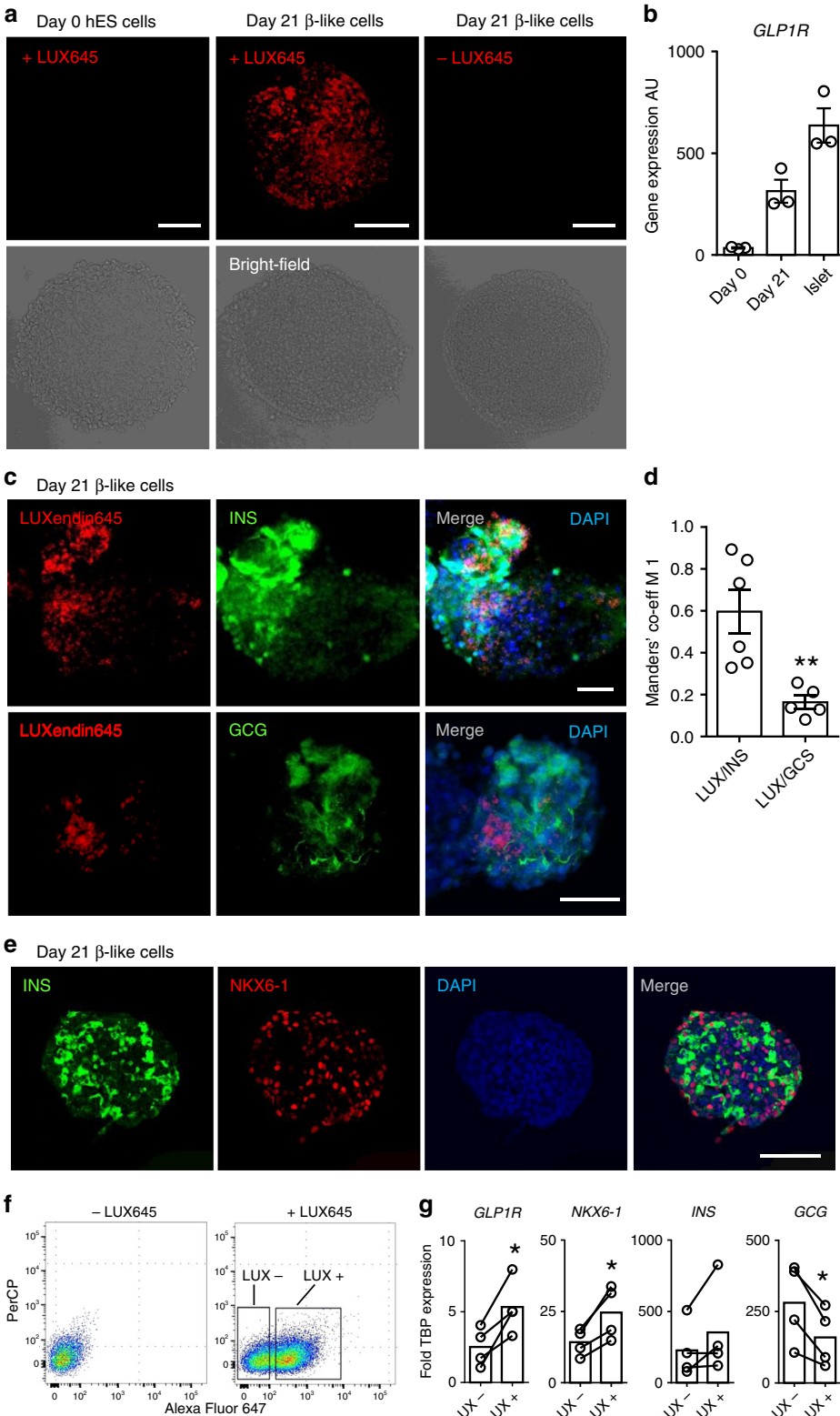

with *Glp1r* forward primer 5′-CAGGCGCTCAGAGCTAGAAGC-3′ and with *Glp1r* wild-type reverse primer 5′-CCAGGGCTCACCTGAGGG-3′ or *Glp1r* knockout 5′-CCAGGGCTCACCTGAGGC-3′ to amplify and detect the WT or mutant allele, respectively. Animals were bred as heterozygous pairs to ensure *Glp1r*[+/+] littermates. *Glp1r*[(GE)−/−] animals are freely available for academic use, subject to a Material Transfer Agreement.

*Ins1Cre*[Thor];*R26*[mT/mG]: To allow identification of β- and non-β-cells, *Ins1Cre*[Thor] animals with Cre knocked-in at the *Ins1* locus (strain B6(Cg)-*Ins1*[tm1.1(cre)Thor]/J; JAX stock no. 026801) were crossed with *R26*[mT/mG] reporter mice (strain B6.129 (Cg)-*Gt(ROSA)26Sor*[tm4(ACTB-tdTomato,-EGFP)Luo]/J; JAX stock no. 007676). Cre-

dependent excision of the floxed allele results in deletion of tdTomato, expression of membrane-localized GFP and thus identification of recombined and non-recombined cells.

*GLU-YFP*: Animals harboring YFP under the control of the glucagon promoter were generated and bred as previously described[55].

*GLP1RCre;LSL-GCaMP3*: To identify GLP1R-expressing cells in the brain, mice with *Glp1r* promoter-drive Cre[7] were bred with stop-flox'd GCaMP3 animals (JAX stock no. 014538).

CD1 wild-type animals were purchased from Charles River Laboratories UK. All studies were performed with 6–12-week-old male and female animals,

**Fig. 8 LUXendin645 labels human ESC-derived β-like cells. a** LUXendin645 (LUX645) labels β-like cells in intact spheroids, which were differentiated and cultured for 21 days. No signal is detected in undifferentiated human ES cells (day 0) or unlabelled β-like cells (-LUX645) (representative images from $n = 6$ spheroids) (scale bar = 100 μm). **b** GLP1R gene expression in day 0 undifferentiated cells, day 21 differentiated β-like cells, and human islets ($n = 3$ donors). **c** LUXendin645 labelling is localized to strongly insulin (INS)-positive but not strongly glucagon (GCG)-positive areas (representative images from $n = 5–6$ spheroids) (scale bar = 50 μm). **d** LUXendin645 (LUX) overlaps more with INS vs. GCG, as calculated using Manders' M1 co-efficient ($n = 5–6$ spheroids) (unpaired Student's $t$-test). **e** Day 21 spheroid sections (5 μm) showing expression of INS and NKX6-1, confirming a differentiated phenotype (representative images from $n = 4$ spheroids) (scale bar = 100 μm). **f** FACS plots of day 21 β-like cells with and without LUXendin645 (LUX645) incubation. LUXendin645+ (LUX+) and LUXendin645− (LUX−) cells were sorted for qPCR. **g** GLP1R, NKX6-1, INS, and GCG gene expression in sorted cells ($n = 4$ spheroids) (connecting bars indicate LUX+ and LUX− populations in the same samples) (paired Student's $t$-test). LUXendin645 was applied at 100 nM. Mean ± s.e.m. are shown. *$P < 0.05$, **$P < 0.01$ for all statistical tests. Source data are provided as a Source Data file.

**Ethical approval**. All animal research complied with the Animals (Scientific Procedures) Act 1986 of the U.K. Approval was granted by the University of Birmingham's Animal Welfare and Ethical Review Body. Procurement of human islets was approved by the Human Research Ethics Board (Pro00013094; Pro00001754) at the University of Alberta and all families of organ donors provided written informed consent. hESC (WA01/H1; hPSCreg name WAe001-A) (obtained from WiCell) were generated by the originating institute with informed consent and ethical approval from the Robert-Koch Institut, Berlin (Az.3.04.02/0101) and NIH (NIHhESC-10-0043). Studies with hESC (WA01/H1) were approved by the BC Children's and Women's Hospital Human Research Ethics Board (Approval #H09-00676). Studies with human tissue were approved by the BC Children's and Women's Hospital Human Research Ethics, University of Birmingham Ethics Committee and the National Research Ethics Committee (REC reference 16/NE/0107, Newcastle and North Tyneside, UK).

**Islet isolation**. Animals were humanely euthanized before injection of collagenase 1 mg/mL (Serva NB8) into the bile duct. Following removal of the inflated pancreas and digestion for 12 min at 37 °C, islets were separated using a Histopaque (Sigma-Aldrich) gradient. Islets were cultured in RPMI medium containing 10% FCS, 100 units/mL penicillin, and 100 μg/mL streptomycin.

**Binding and potency assays**. Binding assays were performed in transiently transfected YFP-AD293-SNAP_GLP1R cells (using PolyJet reagent; SignaGen). Increasing concentrations of compound were applied for 60 min, before imaging using a Zeiss LSM880 meta-confocal microscope configured with GaAsP detectors and ×10/0.45W, ×40/1.00W and ×63/1.20W objectives. YFP, TMR (LUXendin555), and Cy5 (LUXendin645) were excited using $\lambda = 514$ nm, $\lambda = 561$ nm, and $\lambda = 633$ nm lasers, respectively. Emitted signals were captured at $\lambda = 519–574$ nm, $\lambda = 570–641$ nm, and $\lambda = 638–759$ nm for YFP, TMR (LUXendin555), and Cy5 (LUXendin645), respectively. Control experiments were performed in YFP-AD293-SNAP cells, as above.

Potency for cAMP generation and inhibition was tested in heterologous expression systems, comprising either stable CHO-K1-SNAP_GLP1R or HEK-SNAP_GLP1R cells, or transiently transfected YFP-AD293-SNAP_GLP1R cells[21]. Briefly, cells were incubated with increasing concentrations of compound with and without allosteric modulator for 30 min, before harvesting, lysis and measurement of cAMP using either cAMP-Glo™ (Promega) or HTRF (Cisbio) assays, according to the manufacturer's instructions. All assays were performed in the presence of 100–500 μM IBMX to inhibit phosphodiesterase activity. EC$_{50}$ values were calculated using log concentration–response curves fitted with a three-parameter or four-parameter equation.

**Live imaging**. Islets were incubated for 1 h at 37 °C in culture medium supplemented with either 100–250 nM LUXendin555, 50–100 nM LUXendin645, or 100 nM LUXendin651. Islets were imaged using either a Zeiss LSM780 or LSM880 microscope, as above (LUXendin651 was imaged as for LUXendin645). $Ins1Cre^{Thor};R26^{mT/mG}$ islets were excited at $\lambda = 488$ nm (emission, $\lambda = 493–555$ nm) and $\lambda = 561$ nm (emission, $\lambda = 570–624$ nm) for mGFP and tdTomato, respectively. Two-photon imaging of LUXendin645 was performed using a Zeiss LSM 880 NLO equipped with a Spectra-Physics Insight X3 femtosecond-pulsed laser and ×20/1.00W objective. Excitation was performed at $\lambda = 800$ nm and emitted signals detected at $\lambda = 638–759$ nm.

**cAMP imaging**. Islets were transduced with adenovirus harboring the FRET sensor, Epac2-camps[56] (a kind gift from Prof. Dermot Cooper, University of Cambridge), before imaging using a Crest X-Light spinning disk system coupled to a Nikon Ti-E base and ×20/0.4 NA objective. Excitation was delivered at $\lambda = 430–450$ nm using a Lumencor Spectra X light engine. Emitted signals were detected at $\lambda = 460–500$ and $\lambda = 520–550$ nm for Cerulean and Citrine, respectively, using a Photometrics Delta Evolve EM-CCD. Imaging was performed in HEPES–bicarbonate buffer, containing (in mmol/L) 120 NaCl, 4.8 KCl, 24 NaHCO$_3$, 0.5 Na$_2$HPO$_4$, 5 HEPES, 2.5 CaCl$_2$, 1.2 MgCl$_2$, and 3–17 D-glucose. Vehicle (H$_2$O), Exendin4(1–39) (10–20 nM), or Liraglutide (10 nM)

were applied at the indicated time points, with forskolin (10 μM) acting as a positive control.

**Immunostaining**. LUXendin555 (250 nM) or LUXendin645 (50–250 nM) were applied to cells or tissue for 60 min, before fixation in 4% formaldehyde. Primary antibodies were applied overnight at 4 °C in PBS + 0.1% Triton + 1% BSA. Secondary antibodies were applied in the same buffer for 1 h at room temperature, before mounting on slides using Vectashield Hardset containing DAPI. Primary antibodies were mouse monoclonal anti-GLP1R 1:30 (Iowa DHSB; mAb #7F38), rabbit anti-insulin 1:500 (Cell Signaling Technology, #3014), mouse monoclonal anti-glucagon 1:2000 (Sigma-Aldrich, #G2654), and mouse anti-somatostatin 1:5000 (Invitrogen, #14-9751-80). Secondary antibodies were goat anti-mouse DyLight488, goat anti-mouse Alexa Fluor 568, and donkey anti-rabbit DyLight 488 1:1000. Images were captured using an LSM880 meta-confocal microscope. DyLight488 and Alexa Fluor 568 were excited at $\lambda = 488$ nm and $\lambda = 568$ nm, respectively. Emitted signals were detected at $\lambda = 500–550$ nm (DyLight 488) and $\lambda = 519–574$ nm (Alexa Fluor 568). GLP1R surface expression was quantified vs. total GLP1R expression, and normalized against Exendin4(1–39) controls.

**Super-resolution microscopy**. *SRRF:* MIN6 were treated with 250 nM LUXendin645 before live imaging, or fixation and mounting on slides using Vectashield Hardset containing DAPI. Imaging was performed using a Crest X-Light spinning disk system in bypass (widefield) mode. Excitation was delivered at $\lambda = 640/30$ nm through a ×60/1.4 NA objective using a Lumencor SPECTRA X light engine. Emission was collected at $\lambda = 700/75$ nm using a Photometrics Delta Evolve EMCDD. A 250–500 frame raw image sequence was captured (~2 min) before offline super resolution radial fluctuation (SRRF) analysis to generate a single super-resolution snapshot using the NanoJ plugin for ImageJ (NIH)[38].

*STED microscopy:* MIN6 cells were treated with 100, 200, and 400 nM LUXendin651 before fixation (4% paraformaldehyde, 20 min). Cells were mounted in Mowiol supplemented with DABCO and imaged on an Abberior STED 775/595/RESOLFT QUAD scanning microscope (Abberior Instruments GmbH, Germany) equipped with STED lines at $\lambda = 595$ and $\lambda = 775$ nm, excitation lines at $\lambda = 355$, 405, 485, 561, and 640 nm, spectral detection, and a UPlanSApo ×100/1.4 oil immersion objective lens. Following excitation at $\lambda = 640$ nm, fluorescence was acquired in the spectral window $\lambda = 650–800$ nm. For live-imaging, MIN6 cells were seeded on 18 mm coverslips 24–48 h prior to treatment with 400 nM LUXendin651 for 30–45 min before washing once in full medium. Coverslips were transferred into a magnetic chamber (Chamlide CMB, Live Cell Instrument) and washed once with HBSS buffer (Lonza, with additional 5 mM HEPES bubbled with carbogen for 5 min and pH adjusted to 7.4 with NaOH), which was also used as imaging buffer at 24 °C. Live imaging was performed within 45 min after mounting.

Deconvolution was performed with Richardson–Lucy algorithm on Inspector software. FWHM was measured on raw data and calculated using OriginPro 2017 software with Gaussian fitting ($n = 15$ profiles). Minimum and maximum intensity values refer to intensities after deconvolution for STED images and smoothing with a 1-pixel lowpass Gaussian filter for confocal images. Spatial GLP1R expression patterns were analyzed using the $F$- and $G$-functions, where $F =$ distance between an object of interest and its nearest neighbor, and $G =$ distance from a given position to the nearest object of interest (FIJI Spatial Statistic 2D/3D plugin)[57]. Both measures were compared to a random distribution of the same measured objects, with a shift away from the mean ± 95% confidence intervals indicating a non-random or more clustered organization (i.e. more space or smaller distance between objects). Cells possessing highly concentrated GLP1R clusters were identified based upon their fluorescence above a threshold of the population mean fluorescence +1 s.d.

*Single-molecule microscopy:* For single-molecule experiments, CHO-K1-SNAP_GLP1R cells were seeded onto 25 mm clean glass coverslips at a density of $3 \times 10^5$ per well. On the following day, cells were labeled in culture medium with 100 pM LUXendin645 or LUXendin651 for 20 min; this concentration avoids labeling all GLP1R, which would otherwise preclude single-molecule analysis in a stable cell line. At the end of the incubation, cells were washed $3 \times 5$ min in culture medium. Cells were then imaged at 37 °C in phenol-red free Hank's balanced salt

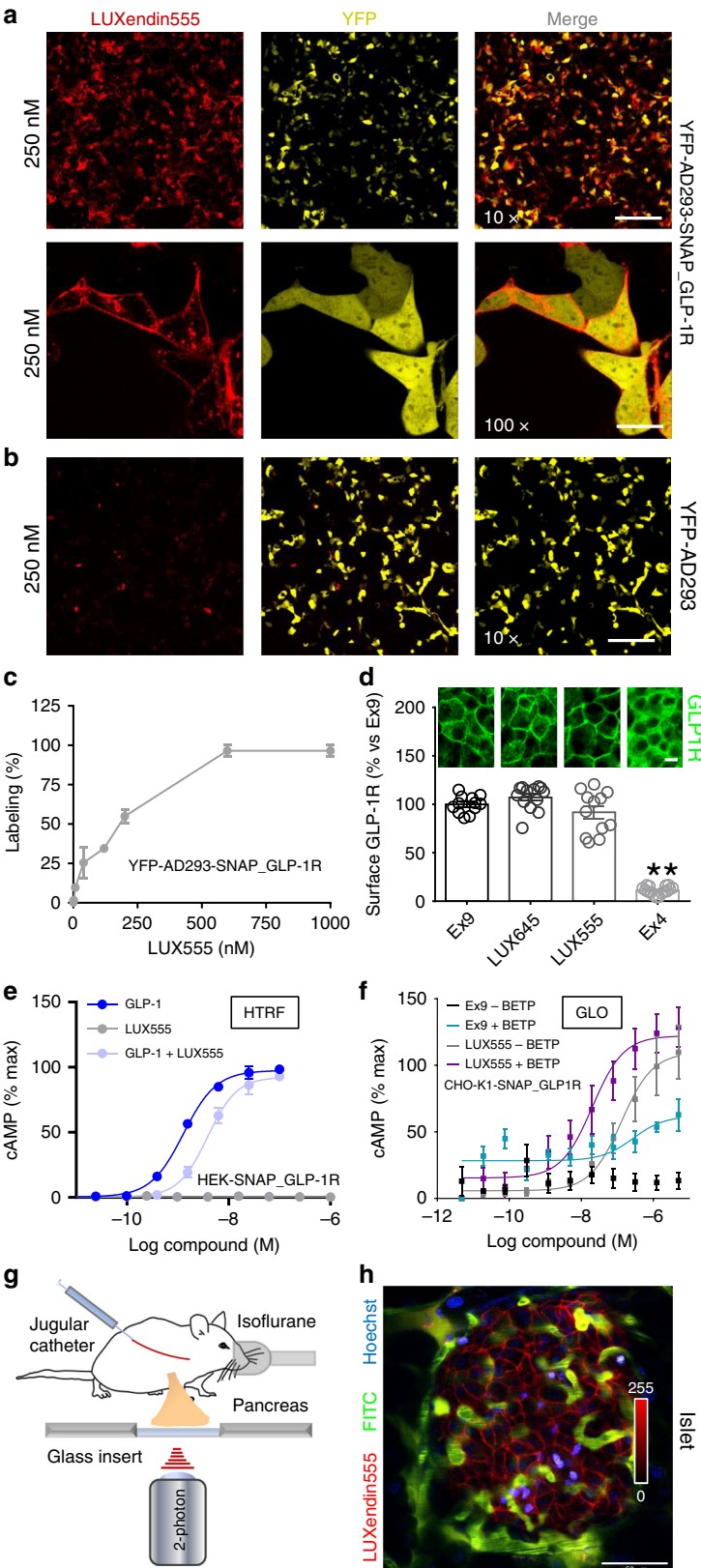

solution, using a custom built TIRF microscope (Cairn Research) based on an Eclipse Ti2 (Nikon, Japan) equipped with an EMCCD camera (iXon Ultra, Andor), 637 nm diode laser, and a ×100 oil-immersion objective (NA 1.49, Nikon). Image sequences were acquired with an exposure time of 60 ms.

Image sequences were analyzed with an automated particle detection software (utrack) in the MATLAB environment[40,58]. To analyze the motion of receptors, the time-averaged mean-squared displacement (TA-MSD)[59] of individual trajectories

from TIRF image sequences was computed[40]. To calculate the diffusion coefficient (D), the TA-MSD data were fitted with the following equation:

$$TA - MSD(t) = 4Dt^\alpha + 4\sigma_{err}^2$$

where $t$ indicates time, $\alpha$ is the anomalous diffusion exponent and $\sigma_{err}$ is a constant offset for localization error. Only trajectories lasting at least 50 frames were

**Fig. 9 LUXendin555 allows in vivo labeling of islets. a–c** LUXendin555 labels YFP-AD293_SNAP-GLP1R **a** but not YFP-AD293 **b** controls with max labeling at 600 nM **c** ($n = 3$ independent assays) (×10 scale bar = 213 μm; ×100 scale bar = 21 μm). **d** Surface GLP1R expression is similar in **LUXendin555** − (LUX555), **LUXendin645**− (LUX645), and 250 nM Exendin4(9–39)-treated islets (100 nM Ex4, +ve control) (representative images shown above each bar) (one-way ANOVA with Bonferroni's test; $F = 173.3$, DF = 3) ($n = 12$ islets, seven animals, three separate islet preparations) (scale bar = 17 μm). **e** **LUXendin555** behaves as an antagonist in HEK-SNAP_GLP1R cells using HTRF-based assays ($n = 4$ independent assays in duplicate). **f** **LUXendin555** displays agonist activity in CHO-K1-SNAP_GLP1R cells, as assessed using luciferase-based detection (GLO) ($n = 3$ independent assays) (positive allosteric modulation was achieved using 25 μM BETP). **g** Schematic depicting the two-photon imaging set up for visualization of the intact pancreas in mice. **h** Representative image showing that **LUXendin555** (100 pmol/g, IV) labels cell membranes in an islet surrounded by the vasculature in vivo ($n = 3$ animals) (scale bar = 50 μm). **LUXendin645** and **LUXendin555** were applied to cells/islets at 100 and 250 nM, respectively. GLP-1 glucagon-like peptide-1; Ex9 Exendin4(9–39); Ex4 Exendin4(1–39). Mean ± s.e.m. are shown. **$P < 0.01$ for all statistical tests. Source data are provided as a Source Data file.

analyzed ($n_{traj} = 5057$ for Cy5 and 8612 for SiR). Trajectories were then categorized according to the diffusion parameters $D$ and $\alpha$. Particles with $D < 0.01$ μm$^2$ s$^{-\alpha}$ were considered to be immobile. Normal diffusion was assigned to particles that had $D \geq 0.01$ μm$^2$ s$^{-\alpha}$ and $0.75 \leq \alpha \leq 1.25$. Sub-diffusion and super-diffusion were assigned to particles with $D \geq 0.01$ μm$^2$ s$^{-\alpha}$ and $\alpha < 0.75$ or $\alpha > 1.25$, respectively.

**Brain labelling**. Mice were injected subcutaneously with 100 pmol/g of **LUXendin645** and left for two hours before terminal anaesthesia and transcardial perfuse fixation with 4% fresh formalin. Brains were serially sectioned at 30 μm and mounted on slides before imaging, as above. Super-resolution snapshots (~140 nm lateral resolution) were acquired using a Zeiss LSM880 equipped with an Airyscan module and a ×63/1.2W objective. Brain clearing was carried out using the 3DISCO protocol[60]. Samples were suspended on a needle, before imaging using a custom-built optical projection tomography (OPT) platform, with images collected after excitation at $\lambda = 470$ and 660 nm. Images were reconstructed using custom-written MATLAB scripts and visualized in Volocity (Perkin-Elmer).

**Two-photon in vivo imaging**. Female and male C57BL/6J mice 7–12 weeks of age were used. Each mouse was anesthetized with isoflurane and a small, 1 cm vertical incision was made at the level of the pancreas. The exposed organ was orientated underneath the animal and pressed against a 50 mm glass-bottom dish for imaging on an inverted microscope. Body temperature was maintained using heat pads and heating elements on the objective. The mouse received Hoechst 33342 (1 mg/kg in PBS) to label nuclei, a 150 kDa fluorescein-conjugated dextran (1 mg/kg in PBS) to label vasculature, and 75 μL of 30 μM **LUXendin555** via retro-orbital IV injection. Images were collected using a Leica SP8 microscope, equipped with a ×25/0.95 NA objective and Spectra Physics MaiTai DeepSee mulitphoton laser. Excitation was delivered at $\lambda = 850$ nm, with signals collected with a HyD detector at $\lambda = 460/50$, $\lambda = 525/50$, $\lambda = 624/40$ nm for Hoechst, FITC, and **LUXendin555**, respectively. Blood was collected from the tail vein prior to and 30 min after **LUXendin555** injection, and glucose was measured using an AlphaTrak2 glucometer. All in vivo imaging experiments were performed with approval and oversight from the Indiana University Institutional Animal Care and Use Committee (IACUC).

**Stem cell differentiation and gene expression analyses**. WA01/H1 hESCs were differentiated using the protocol published by Nair et al.[61]. Briefly, dissociated H1 hESCs were plated on six-well plates at a density of 5.5 million cells in 5.5 mL media per well. The plates were incubated at 37 °C and 5% CO$_2$ on an orbital shaker at 100 rpm to induce spheroid formation. After 24 h, six-step differentiation was induced. Differentiation was stopped at day 21 and spheroids labelled with **LUXendin645**, fixed in 4% formaldehyde and co-stained with insulin and glucagon. To confirm differentiation, a subset of spheroids were paraffin-embedded, sectioned at 5 μm and stained for insulin and NKX6-1. Primary antibodies were guinea pig anti-insulin 1:500 (Dako Cytomation, #A0564), mouse monoclonal anti-glucagon 1:2000 (Sigma-Aldrich, #G2654), and rabbit monoclonal anti-NKX6-1 (D8O4R) 1:400 (Cell Signaling, #54551). Secondary antibodies were donkey anti-guinea pig FITC, donkey anti-mouse FITC, and donkey anti-rabbit Cy3 1:200 (Jackson Immuno Research Laboratories, #706-096-148, #715-096-150, #711-166-152). The spheroids and sections were imaged using a Leica SP8 confocal microscope with a ×20/0.75 IMM objective. Manders' co-efficient (Coloc 2 plugin for FIJI) was used to quantify the extent of overlap of **LUXendin645** signal with the insulin or glucagon channels (M1).

nCounter gene expression assay (Nanostring, WA) was used to assess *GLP1R* gene expression in hESCs, differentiated β-like cells, and human islets. The values are normalized to six different housekeeping genes (*B2M*, *GAPDH*, *GUSB*, *HPRT1*, *POLR2A*, and *TBP*). Human islets were obtained from Alberta Islet Distribution Program. Sex, age, and BMI of each islet sample were: #1: Male, 53 y.o., 33.7 kg/m$^2$, #2: Female, 17 y.o., 23.4 kg/m$^2$, #3: Male, 18 y.o., 22 kg/m$^2$. WA01/H1 hESCs were obtained under MTA from WiCell (Madison WI) and institutional use was approved by the BCCHR/UBC Human Research Ethics Board (Approval # H09-00676).

For FACS analysis, **LUXendin**-labelled spheroids were collected, incubated with Accumax at 37 °C for 10 min and dissociated into single cells. CMRL with 1% BSA was added, followed by filtering through a 40-mm nylon filter. Cells were

centrifuged for 5 min at $200\times g$, washed with PBS, and resuspended in 500 μL of PBS. **LUXendin+** and **LUXendin−** cells were sorted into TRIzol for further qPCR using a BD FACSAria IIu. RNA was isolated with TRIzol, DNase-treated with Turbo DNASe Free, and reverse transcribed with Superscript III. TaqMan qPCR was performed and data were normalized by *TBP*. Primers used were: *GLP1R* (forward [F], 5′-GTGCTATACATCCACTTCAGGG-3′; reverse [R], 5′-GCTCTG GTTATCGCCTCTG-3′; and probe 5-TCCACCTGAACCTGTTTGCATCCT-3′), *NKX6-1* (F, 5′-TCGTTTGGCCTATTCGTTGG-3′; R, 5′-TGTCTCCGAGTCCTG CTTC-3′; and probe 5-TGCTTCTTCCTCCACTTGGTCCG-3′), *INS* (F, 5′-CTA GTGTGCGGGGAACG-3′; R, 5′-CACGCTTCTGCAGGGAC-3′; and probe 5-C GGCGGGTCTTGGGTGTGTA-3′), *GCG* (F, 5′-GTCCAGATACTTGCTGTAGT CAC-3′; R, 5′-ACGTTCCCTTCAAGACACAG-3′; and probe 5-ATGGCGCTTGT CCTCGTTCATCT-3′), and *TBP* (F, 5′-GAGAGTTCTGGGATTGTACCG-3′; R, 5′-ATCCTCATGATTACCGCAGC-3′; and probe 5-TGGGATTATATTCGGCG TTTCGGGC-3′).

**Statistical analyses**. Measurements were performed on discrete samples unless otherwise stated. All analyses were conducted using GraphPad Prism software. Unpaired or paired Student's t-test was used for pairwise comparisons. Multiple interactions were determined using one-way or two-way ANOVA followed by Bonferroni's, Dunn's, or Sidak's posthoc tests (accounting for degrees of freedom).

**Reporting summary**. Further information on research design is available in the Nature Research Reporting Summary linked to this article.

## Data availability
**LUXendins** are freely available for academic use. *Glp1r*$^{(GE)-/-}$, *GLU-YFP*, and *GLP1RCre* animals are subject to a Material Transfer Agreement. The source data underlying Figs. 2a–c, g, k, 3b–f, 4f and h, 5e–g, 6c, 8b and d, and 9c–f and Supplementary Figs. 12, 15, and 16 are provided as a Source Data file. Raw image files are available upon reasonable request.

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

## Acknowledgements

We thank Bettina Mathes and Alexandra Teslenko for excellent synthetic support. D.J.H. was supported by a Diabetes UK R.D. Lawrence (12/0004431) Fellowship, a Wellcome Trust Institutional Support Award, MRC Confidence in Concept, MRC (MR/N00275X/1 and MR/S025618/1) Project and Diabetes UK (17/0005681) Project Grants. A.T. and B.J.J. were funded by an MRC Project Grant (MR/R010676/1). G.D. was supported by an MRC CDA Fellowship (MR/P009824/1). A.K.L. was supported by R03 DK115990 and Human Islet Research Network UC4 DK104162 (to A.K.L.; RRID: SCR_014393). Intravital microscopy core services were supported by NIH NIDDK Grant P30 DK097512 to the Indiana University School of Medicine. F.C.L. received operating support from CIHR PJT 156377. Fellowship support was provided by the Juvenile Diabetes Research Foundation (S.S.), and the Michael Smith Foundation for Health Research (S.S.), and the Manpei Suzuki Diabetes Foundation (S.S.). S.T. was supported by grants from the MRC (MR/N02589X/1) and NIH (R01 DK095757). B.H.

was supported by the Wellcome Trust (095101, 200837 and 106130). D.C. was funded by the Deutsche Forschungsgemeinschaft (SFB/Transregio 166-Project C1) and a Wellcome Trust Senior Research Fellowship (212313/Z/18/Z). This project has received funding from the European Research Council (ERC) under the European Union's Horizon 2020 research and innovation programme (Starting Grant 715884 to D.J.H.). We thank Prof. Anna Gloyn (University of Oxford) for provision of reagents, Dr. Birgit Koch (MPI, Heidelberg) for helpful discussions on $Glp1r^{(GE)-/-}$ mice and Dr. Jacqueline Naylor (MedImmune) for generation of parental SNAP_GLP1R-INS1$^{GLP1R-/-}$ cells. Lastly, we thank Dr. Jocelyn E. Manning Fox and Prof. Patrick E. MacDonald for provision of human islets via the Alberta Diabetes Institute IsletCore at the University of Alberta in Edmonton with the assistance of the Human Organ Procurement and Exchange (HOPE) program, Trillium Gift of Life Network (TGLN) and other Canadian organ procurement organizations.

## Author contributions

J.A., K.J., T.P., J.B. and D.J.H. devised the studies. J.A., A.A., D.N., N.H.F.F., F.B.A., Z.S., B.H., B.J.J., M.A.L., D.I.B., A.T., G.D., S.T., T.P., J.B. and D.J.H. performed experiments and analyzed data. J.A. and A.B. generated mice. F.R. generated and provided reporter mice. J.A., E.D'E., J.B. and D.J.H. performed super-resolution imaging. C.A.R. and A.K.L. performed in vivo imaging experiments. Z.K. and D.C. performed and analyzed single-molecule microscopy experiments. S.S. and F.C.L. performed and analyzed experiments with hESC. J.A., S.T., K.J., T.P., J.B. and D.J.H. supervised the work. J.A., T.P., J.B. and D.J.H. wrote the manuscript with input from all the authors.

## Competing interests

The authors declare no competing interests.
