## [Peer Review File · Nature Communications]

Reviewers' comments:

Reviewer #1 (Remarks to the Author):

The work of Ast, Hodson and co-worker present a new far red fluorescent label specific for glucagon-like peptide-1 receptor GLP1R.

They have synthesized three LUXendin555, LUXendin651 and LUXendin645 which resulted compatible with several imaging methods across scales from confocal and Two Photon microscopy, where they show proper tissue penetration of the labels to super resolution imaging where STED images show receptors organizing into clusters.

Overall the work is well done presenting a plethora of possible applications compatible with different imaging modalities. Also, the specificity of LUXendin is well demonstrated.

The visualization of fine molecular organizations calls for the development of smaller yet specific labels compatible with super resolution imaging methods. LUXendin has the potential to extend our understanding of GLP1R molecular dynamics therefore I recommend the publication of this work in Nature Communication. To improve the clarity of the manuscript for a broader audience of imaging and biological experts, I suggest the authors to work on the points stated below:

1. Figure 5 legend. The authors mentioned an axial resolution of 50 nm but not further information or data are provided. Is the image recorded with a 3D STED scheme? Did they measure an effective cluster along the optical axis of 50 nm?

Further proves are needed to support this statement. An xz image of LUXendin651-treated MIN6 cells would strength the observation.

Typically, commercial 3D STED system can reach a spatial resolution along the optical axis around 70-100 nm at best. If the images show 50 nm spatial improvements it deserves to be reported and properly quantified.

2. Within the beneficial properties of LUXendin651 is the possibility to label living cells. However, the super resolution imaging was done on fixed specimens.

Is there a specific reason/limitation for SRRF analysis or STED in living cells, were membranes should be better preserve? I believe that the dynamic behavior of the receptor can hamper super resolution imaging in large field of view, but in small area it should be possible to prove the existence of nanodomains.

3. It would help to have further comparison between confocal and STED imaging. For example, adding the zoom of the confocal image side by side to the STED zoom image in Figure 5 will help to show the improvement in recorded spatial details.

All the images lack an intensity bar, especially in the super resolution data it would provide important information on the signal level. Please add them.

Reviewer #2 (Remarks to the Author):

This paper provides novel ligands to assess GLP-1R expression in native systems. In particular they have demonstrated staining of the GLP-1R in islets with greater sensitivity than currently available tools, and the utility of these could be used to assess expression in other tissues, providing a tool for the community. The studies are very elegant and the specific fluorophores used provide very good sensitivity. The superresolution and STED data are extremely interesting revealing clustering of receptors in different microdomains within the cell membrane. In addition, the ability to use these types of ligands to label receptors in vivo is very powerful. However, I do have a number of questions and concerns regarding the interpretation for some of the data and how the particular ligands that have been developed will be utilised further to understand GLP-1R function.

In this study 3 ligands were described LUXendin555 (555), LUXendin645 (645), LUXendin651 (651) with 555 being described as an agonist for both cAMP and internalisation, whereas 645 and

651 were described as antagonists. While 645 has been profiled in cAMP and does antagonise the action of GLP-1 (fig 2a), Fig 2b looks like there is some partial agonist behaviour of 645 (about 20 % of the forskolin response). This is a similar degree of agonism to BETP (alone) which is a known PAM agonist. In a competition assay, a partial would still antagonise the response of a full agonist to give the profile observed in Fig2A. I would recommend increasing the N numbers for these data (currently n=3) to determine how robust this data is. Currently there is a large degree of error in the data for fig2b that supports 645 is not an antagonist but rather a partial agonist. Partial agonist activity of 645 is also supported by the imaging data where there are punctate spots inside the cell suggesting some receptor internalisation from the ligand, but to a lesser degree than a full agonist such as 555 or Ex4. Distinguishing whether this ligand is a weak partial agonist as it appears or truly an antagonist is an important distinction to be able to make as this affects the interpretation of the data in the remainder of the study. Moreover, in these experiments (Fig 2b), Exendin-4 is used in parallel as the control and this has a much lower potency than I would expect given that its affinity for the receptor is nM. This ligand should give a similar (or slightly higher) potency to GLP-1 (shown in panel A).

The STED data is really interesting clearly showing the first direct evidence using super resolution that GLP-1Rs are clustered in hotspots, likely organised in nanodomains. However, I think the authors need to be more cautious with their interpretation that what they are studying is non-stimulated cells. The receptors contain a ligand and these ligands have not been extensively characterised for their full pharmacological behaviour (I couldn't see any direct data for 651 at all in this paper that needs to be addressed). There is multiple lines of evidence that GPCR ligands can promote redistribution of receptors within the cell membrane and that these redistributions differ depending on the nature of the ligand. In addition, there is evidence (<https://www.biorxiv.org/content/10.1101/492496v1>) that the GLP-1R can redistribute into cholesterol rich domains within the membrane and this is reported to be required for receptor internalisation, with the extent of GLP-1R redistribution and internalisation observed dependent on the activating ligand. This should be considered in the discussion of the STED data as, while the data is very nice and interesting, it is unclear if the distributions of the receptor within the membrane are ligand-induced or if this is how receptors are ordered in the cells in the absence of ligand. The STED experiments for this part of the study were performed with the 651 ligand and there is no pharmacology supporting whether this is an antagonist or an agonist in any assay. Given changing the probe in 555 induced greater agonism than 645, it is likely that the SiR probe may also have differing pharmacology to 645. In addition, the microscopy data with 645 and 651 clearly shows punctate spots that are not at the plasma membrane suggesting these ligand are capable of internalising the receptor to some extent.

Line 134- Maximum labelling occurs at 100nM is not consistent with the figures that suggest maximal labelled was observed at 500nM (The KA (50 % labelling) occurs at around 100nM).

Line 241 suggests that unlike the LUX agonists Exendin-4 is not modulated by BETP. This is not technically true. While no modulation is observed in cAMP exendin-4 activity can be modulated when assessing other pathways that can be activated by the GLP-1R such as pERK1/2 and Ca²⁺ (Mol.Pharm Apr; 83(4):822-34).

Line 237 – I would remove the work potentially. 100nM is not potent for peptide ligands at this receptor (in fact, according to the curve fit in fig 2b, 645 has a similar potency, but much lower maximum response (efficacy)).

Line 267 – How does the use of a peptide antagonist encourage receptor recycling when it does not cross the membrane?

Line 269 – see comment above. I am not convinced by the data provided that the probes developed are truly antagonists. Additional data would need to be provided to confirm this.

Line 273-275 – Due to the probe-dependent nature of PAMs, the expression and trafficking of native receptors in the presence of PAMs will differ depending on the nature of the orthosteric ligand. While the probes could be used to look at how these specific LUXendin ligands influence receptor expression and distribution in the presence of a PAM, each specific ligand of interest

would need to be developed with the specific fluorescent probes (with the probe not altering its pharmacology) to really be useful to understand receptor trafficking in a meaningful manner. This should be included in the discussion.

Line 323-327 – I don't agree that the C-terminal rhodamine counteracts the removal of the first 8 residues as this suggests 555 is as good an agonist as Exendin-4 and it isn't. It is also likely to have a different overall signalling profile. It is also incorrect to state that physiologically mediated DPPIV removes the first 8 residues. Exendin-4 is resistant to DPPIV cleavage and DPPIV only cleaves the first 2 amino acids of GLP-1(7-36) that is the predominant and high affinity form of GLP-1 and the first two amino acids of oxyntomodulin that is also an endogenous ligand.

Line 329-331 -What data supports that LUXendin555 maintains stable glycaemia?

Methods:

Why were different concentrations of ligands used for the different assays (where single concentrations were assessed)? This should be justified. In addition concentrations of BETP used should also be added to the methods section and figure legends.

Generally the number of individual n numbers for experiments are low. For data that are reported from islets (as number of islets), were these derived from distinct preparations or from one islet isolation? Where cells are reported (ie Figure 2 m n=4-5 images, 57-64 cells), how many individual experiments were performed? This information should be included for each experiment as it speaks to reproducibility. The two-photon in vivo imaging was from 1 experiment on 1 mouse. Is this procedure reproducible?

Line 412-418 - Were cAMP assays performed in the presence or absence of IBMX?

Line 430 – Which cAMP EPAC2 sensor was used or these studies?

Line 453 – What concentration of Luxendin was used here?

Figure 1: Peptide sequences do not align to each other (nor does the numbering).

Figures 2 and 6: For the cell surface expression, what are the data normalised to? If this is total GLP-1R expression, what was used for the negative control? Figure 2g and 6d both report on 645, but in fig2 this is around 70 % expression and in 6d 90%. What concentration ligand was used for each? It would be useful to include all ligand concentrations in either the figure or the legend.

Reviewer #3 (Remarks to the Author):

The article by Ast et al., "LUXendins reveal endogenous glucagon-like peptide-1 receptor distribution and dynamics", is well-written and the studies presented in the report appear to be of high quality. The key finding is the discovery of far-red fluorescent-containing GLP1R probes that may offer significant advantages over current methods to assess GLP1R expression that use either genetic approaches, or antibody staining/labelled agonist probes. The report clearly presents POC data using pancreatic islets by showing strong membrane labeling for GLP1R, and the authors nicely show loss of the signal in islets from Glp1r knockout mice (as would be expected). A potential advance for the field is the visualization of membrane nano-domains and the ability to track single receptor subpopulations.

While these tools appear to have great promise, detection of the GLP1R in pancreatic islets is a minimal bar, and the paper would be significantly improved if the technology was applied to characterizing GLP1R expression in the peripheral and central nervous systems and throughout the gastrointestinal tract. It is still not completely understood in GLP1R biology how activating there receptor in the periphery and brain impact glucose and energy metabolism. Having the Glp1r knockout mice to use as a negative control would allow exploring the whole animal for the key

sites of GLP1 action.

The authors show these probes can be used to assess GLP1R internalization. Therefore, a potential utility of these probes would be to assess GLP1R internalization following treatment with a short-acting versus long-acting GLP1R agonist. Demonstrating differences in GLP1R internalization in animals treated with these types of therapeutic agents could provide insight into what the most desirable profile of a GLP1 medicine would be.

The finding that different fluorophores confer agonistic behavior on the LUX backbone does not indicate better/stabilized incretin therapeutics can be developed. This comment is over-stated and should be removed.

We would like to thank the three expert Reviewers and Editor for their constructive and helpful feedback. We have taken on board the Reviewers' comments and revised the manuscript accordingly. In particular, we have: 1) re-performed cAMP concentration-response experiments to verify ligand properties; 2) shown intense membrane-localized **LUXendin** labelling in the brain following simple subcutaneous injection; 3) described the existence of GLP1R nanodomains in forebrain and hindbrain neurons, as well as the choroid plexus; 4) shown **LUXendin**-labeling during differentiation of hESC-derived β -like cells; 5) increased n numbers where relevant and included the number of independent replicates; and 6) inserted numerous caveats into the results/discussion. Lastly, we noticed a small error in the quantification of GLP1R+ non-beta cells, which was updated in the online preprint whilst the studies were under review. Using the correct denominator, 6% of alpha cells are found to express GLP1R, which still represents a 5-fold increase in sensitivity over similar fluorophore-agonist or antibody approaches, but is similar to that detected with *GLP1RCre* reporter animals.

As a consequence of the revisions, we have added a number of co-authors, including: Giuseppe D'Agostino (University of Manchester), Maria A. Lucey (Imperial College London), Dan Brierley (UCL), and Francis C. Lynn and Shugo Sasaki (both University of British Columbia), who between them performed brain and hESC studies. We have also included Frank Reimann (University of Cambridge) who provided *GLU-YFP* and *GLP1RCre;LSL-GCaMP3* reporter mice.

With these changes, we feel that the utility and specificity of **LUXendins** has been further demonstrated, with benefit for the wider metabolism and now neuroscience and stem cell research fields. We therefore hope that you will find the manuscript acceptable for publication in *Nature Communications*.

Reviewer #1:

The work of Ast, Hodson and co-worker present a new far red fluorescent label specific for glucagon-like peptide-1 receptor GLP1R.

They have synthesized three LUXendin555, LUXendin651 and LUXendin645 which resulted compatible with several imaging methods across scales from confocal and Two Photon microscopy, where they show proper tissue penetration of the labels to super resolution imaging where STED images show receptors organizing into clusters.

Overall the work is well done presenting a plethora of possible applications compatible with different imaging modalities. Also, the specificity of LUXendin is well demonstrated.

The visualization of fine molecular organizations calls for the development of smaller yet specific labels compatible with super resolution imaging methods. LUXendin has the potential to extend our understanding of GLP1R molecular dynamics therefore I recommend the publication of this work in Nature Communication. To improve the clarity of the manuscript for a broader audience of imaging and biological experts, I suggest the authors to work on the points stated below.

- We are pleased that the reviewer liked the manuscript and appreciates the flexibility of the approach.

Figure 5 legend. The authors mentioned an axial resolution of 50 nm but not further information or data are provided. Is the image recorded with a 3D STED scheme? Did they measure an effective cluster along the optical axis of 50 nm?

Further proves are needed to support this statement. An xz image of LUXendin651-treated MIN6 cells would strength the observation.

Typically, commercial 3D STED system can reach a spatial resolution along the optical axis around 70-100 nm at best. If the images show 50 nm spatial improvements it deserves to be reported and properly quantified.

- The stated value was in fact the lateral and not axial resolution of the representative image. We thank the reviewer for spotting this error on our behalf. The figure legend has been modified as follows:

“**b-d** Confocal and STED snapshots of endogenous GLP1R in **LUXendin651**-treated MIN6 cells at FWHM = 70 ± 10 nm. Note the presence of punctate GLP1R expression as well as aggregation/clustering in cells imaged just away from (**b**), close to (**c**) or next to (**d**) the coverslip using STED microscopy (n = 8 images from three independent repeats) (scale bar = 2 μ m).”

Within the beneficial properties of LUXendin651 is the possibility to label living cells. However, the super resolution imaging was done on fixed specimens.

Is there a specific reason/limitation for SRRF analysis or STED in living cells, were membranes should be better preserve? I believe that the dynamic behavior of the receptor can hamper super resolution imaging in large field of view, but in small area it should be possible to prove the existence of nanodomains.

- We have now extended super-resolution imaging to living MIN6 cells using SRRF and STED, further showing the utility of **LUXendins** for understanding higher GLP1R organization (Figure 5h and i). STED imaging was possible and clearly improved resolution in living cells. Nevertheless, as already pointed out by the reviewer, the visualization of GLP1R nanodomains was more difficult and their concentration sparser, due to the fast dynamics of the receptors (measured in Figure 6). The results discussed as follows:

“**LUXendin651** even allowed GLP1R to be imaged in living MIN6 cells using SRRF and STED, although nanodomains were more difficult to resolve due to the lateral diffusion of receptors (Fig. 5h and i).”

It would help to have further comparison between confocal and STED imaging. For example, adding the zoom of the confocal image side by side to the STED zoom image in Figure 5 will help to show the improvement in recorded spatial details.

- We thank the reviewer for this suggestion, which we have included in the revised manuscript (Figure 5b-d and Figure 5i).

All the images lack an intensity bar, especially in the super resolution data it would provide important information on the signal level. Please add them.

- We have now added **LUXendin** intensity bars to the super-resolution data where photon flux can be limiting.

Reviewer #2:

This paper provides novel ligands to assess GLP-1R expression in native systems. In particular they have demonstrated staining of the GLP-1R in islets with greater sensitivity than currently available tools, and the utility of these could be used to assess expression in other tissues, providing a tool for the community. The studies are very elegant and the specific fluorophores used provide very good sensitivity. The superresolution and STED data are extremely interesting revealing clustering of receptors in different microdomains within the cell membrane. In addition, the ability to use these types of ligands to label receptors in vivo is very powerful. However, I do have a number of questions and concerns regarding the interpretation for some of the data and how the particular ligands that have been developed will be utilised further to understand GLP-1R function.

- We thank the reviewer for their kind words and are pleased that he/she found the studies to be elegant. Based on the helpful feedback, we have worked hard to improve data interpretation, as well as ligand validation.

In this study 3 ligands were described LUXendin555 (555), LUXendin645 (645), LUXendin651 (651) with 555 being described as an agonist for both cAMP and internalisation, whereas 645 and 651 were described as antagonists. While 645 has been profiled in cAMP and does antagonise the action of GLP-1 (fig 2a), Fig 2b looks like there is some partial agonist behaviour of 645 (about 20 % of the forskolin response). This is a similar degree of agonism to BETP (alone) which is a known PAM agonist. In a competition assay, a partial would still antagonise the response of a full agonist to give the profile observed in Fig2A. I would recommend increasing the N numbers for these data (currently n=3) to determine how robust this data is. Currently there is a large degree of error in the data for fig2b that supports 645 is not an antagonist but rather a partial agonist. Partial agonist activity of 645 is also supported by the imaging data where there are punctate spots inside the cell suggesting some receptor internalisation from the ligand, but to a lesser degree than a full agonist such as 555 or Ex4. Distinguishing whether this ligand is a weak partial agonist as it appears or truly an antagonist is an important distinction to be able to make as this affects the interpretation of the data in the remainder of the study. Moreover, in these experiments (Fig 2b), Exendin-4 is used in parallel as the control and this has a much lower potency than I would expect given that its affinity for the receptor is nM. This ligand should give a similar (or slightly higher) potency to GLP-1 (shown in panel A).

- Determining compound cAMP responses proved frustrating, since the fluorescent moiety interfered with our robust LANCE-FRET assay, meaning we had to instead use luminescence readouts. We however agree with the reviewer that it is important to truly know whether the probes are antagonists or partial agonists, since this affects interpretation of the study. We have therefore optimized HTRF cAMP assays using an amplified HEK293-SNAP_GLP1R cell system, which allows us to dilute lysate 100-fold before running the samples. Using these assays, we show that **LUXendin645** and **LUXendin651** do not lead to cAMP generation and display antagonist properties at the GLP1R, with no evidence of weak or partial agonism (Figure S1) (lines 142-145, 223-224).

We also show with the same assay that **LUXendin555** is an antagonist, in contrast to our previous luciferase-based readouts showing weak agonist activity (Figure 8e and f) (lines 291-292). We are unsure about the exact reason for this difference, but it might reflect interference of the TMR moiety with cAMP assay readout. Thus, we would prefer to remain cautious with our interpretation of **LUXendin555** pharmacology and provide the reader with both datasets (Figure 9e and f) accompanied by clear discussion, as follows (lines 424-431):

“**LUXendin555** showed antagonist activity in terms of cAMP generation using a HTRF approach. However, we could not confirm this result using a luciferase-based detection method. The reasons for this are unclear, but might include interference of the red fluorescent TMR with either assay, or alternatively different GLP1R coupling strength between cell lines. As such, use of **LUXendin555** should consider the possibility that the ligand is an antagonist or agonist. Nonetheless, **LUXendin555** retained GLP1R at the membrane, and possesses advantageous properties for *in vivo* imaging including good two-photon cross-section and high quantum yield.”

Lastly, the punctate staining observed in the cytoplasm is likely to be cleaved fluorophore rather than internalized GLP1R, since: 1) fluorophore can be liberated at the cell surface following maleimide exchange with other reactive thiols;¹ and 2) puncta were not apparent in the same samples co-stained with GLP1R mAb (Figure 2j, Figure 2i). We have discussed this in the revised manuscript as follows (lines 341-348):

“In some experiments, we also noticed the presence of punctate **LUXendin645** and **LUXendin651** labelling. Suggesting that this staining pattern reflects cleaved fluorophore rather than internalized GLP1R are the following observations: 1) maleimide exchange with reactive thiols can lead to linker loss,¹ allowing free fluorophore to cross the membrane and accumulate in organelles; and 2) puncta were not apparent in the same samples co-stained with GLP1R mAb.”

The STED data is really interesting clearly showing the first direct evidence using super resolution that GLP-1Rs are clustered in hotspots, likely organised in nanodomains. However, I think the authors need to be more cautious with their interpretation that what they are studying is non-stimulated cells. The receptors contain a ligand and these ligands have not been extensively characterised for their full pharmacological behaviour (I couldn't see

any direct data for 651 at all in this paper that needs to be addressed). There are multiple lines of evidence that GPCR ligands can promote redistribution of receptors within the cell membrane and that these redistributions differ depending on the nature of the ligand. In addition, there is evidence (<https://www.biorxiv.org/content/10.1101/492496v1>) that the GLP-1R can redistribute into cholesterol rich domains within the membrane and this is reported to be required for receptor internalisation, with the extent of GLP-1R redistribution and internalisation observed dependent on the activating ligand. This should be considered in the discussion of the STED data as, while the data is very nice and interesting, it is unclear if the distributions of the receptor within the membrane are ligand-induced or if this is how receptors are ordered in the cells in the absence of ligand. The STED experiments for this part of the study were performed with the 651 ligand and there is no pharmacology supporting whether this is an antagonist or an agonist in any assay. Given changing the probe in 555 induced greater agonism than 645, it is likely that the SiR probe may also have differing pharmacology to 645.

- The reviewer is correct that GLP1R redistribution at the membrane is likely to be ligand-dependent and that we may not necessarily be imaging the non-stimulated state. To address this comment, we have: 1) provided full pharmacological validation of **LUXendin651** showing antagonist behavior (see above); 2) stated that GLP1R nanodomains were imaged in the presence of **LUXendin651**; 3) discussed the need to repeat experiments with different agonists/antagonists to assess the ligand-dependency of GLP1R redistribution and trafficking; and 4) mentioned the utility of **LUXendin651** for the production of other fluorescent ligands that would allow super-resolution examination of nanodomain architecture in response to different activation modes. The modified paragraph is below (lines 406-413):

“STED imaging showed that endogenous GLP1R possess a higher structural order in the presence of **LUXendin651** binding: namely organization into nanodomains at the cell membrane. Since GLP1R plasma membrane distribution is ligand-dependent, for example via effects on palmitoylation and clustering into cholesterol-rich nanodomains² it will be interesting to repeat these experiments using a range of agonists/antagonists. Indeed, **LUXendins** provide an ideal template for the production of fluorescent ligands that would allow super-resolution examination of nanodomain architecture in response to different activation modes.”

Moreover, we have analyzed both **LUXendin645** and **LUXendin651** SMLM movies to show that GLP1R diffusion profiles are remarkably similar in the presence of either ligand (Figure 6c) (lines 248-249). Together with our extensive pharmacological work-up, these data suggest that installation of SiR does not unduly affect ligand behavior.

The microscopy data with 645 and 651 clearly shows punctate spots that are not at the plasma membrane suggesting these ligand are capable of internalising the receptor to some extent.

- Punctate spots upon confocal imaging are likely to be cleaved fluorophore (please see above). For super-resolution microscopy, imaging was performed in three different

regions spanning next to, close to and just away from the coverslip. Thus, at the axial resolutions employed here, which is still diffraction limited, the punctate spots represent GLP1R at the membrane. We have inserted new images to better show an increase in punctate GLP1R staining with proximity to the coverslip (Fig. 5b-d), where a larger surface area of the membrane is visualized.

Line 134- Maximum labelling occurs at 100nM is not consistent with the figures that suggest maximal labelled was observed at 500nM (The KA (50 % labelling) occurs at around 100nM).

- We apologize for this error- maximum labeling was seen at 250-500 nM. This is now corrected in the revised manuscript.

Line 241 suggests that unlike the LUX agonists Exendin-4 is not modulated by BETP. This is not technically true. While no modulation is observed in cAMP exendin-4 activity can be modulated when assessing other pathways that can be activated by the GLP-1R such as pERK1/2 and Ca²⁺ (Mol.Pharm Apr;83(4):822-34).

- We agree and have removed this sentence.

Line 237 – I would remove the word potentially. 100nM is not potent for peptide ligands at this receptor (in fact, according to the curve fit in fig 2b, 645 has a similar potency, but much lower maximum response(efficacy)).

- We agree and have removed the word 'potent'.

Line 267 – How does the use of a peptide antagonist encourage receptor recycling when it does not cross the membrane?

- Thanks for picking this up. The sentence has been modified as follows:

“Firstly, the use of an antagonist retains more receptor at the cell surface, which likely increases detection capability.....”

Line 269 – see comment above. I am not convinced by the data provided that the probes developed are truly antagonists. Additional data would need to be provided to confirm this.

- Please see the response above. We have now repeated cAMP concentration-response studies and show that the probes possess antagonist behavior.

Line 273-275 – Due to the probe-dependent nature of PAMs, the expression and trafficking of native receptors in the presence of PAMs will differ depending on the nature of the orthosteric ligand. While the probes could be used to look at how these specific LUXendin ligands influence receptor expression and distribution in the presence of a PAM, each specific ligand of interest would need to be developed with the specific fluorescent probes (with the probe not altering its pharmacology) to really be useful to understand receptor trafficking in a meaningful manner. This should be included in the discussion.

- The reviewer makes a good point. We have modified this sentence as follows (lines 338-341):

“Together, these desirable properties open up the possibility to image expression and distribution of native GLP1R over extended periods of time using multiple imaging modalities.”

We have also inserted a caveat into the discussion to mention that our results show trafficking in the presence of **LUXendin** + PAM, and that further studies are required with different pharmacophores to understand whether or not this is ligand-dependent (lines 325-328):

“While **LUXendins** also allowed GLP1R trafficking to be monitored, this required the presence of a PAM to allosterically activate the receptor. Due to the probe-dependent nature of PAMs, **LUXendins** with a number of different pharmacophores would need to be generated to fully assess the ligand-dependency of GLP1R trafficking.”

Line 323-327 – I don't agree that the C-terminal rhodamine counteracts the removal of the first 8 residues as this suggests 555 is as good an agonist as Exendin-4 and it isn't. It is also likely to have a different overall signalling profile. It is also incorrect to state that physiologically mediated DPPIV removes the first 8 residues. Exendin-4 is resistant to DPPIV cleavage and DPPIV only cleaves the first 2 amino acids of GLP-1(7-36) that is the predominant and high affinity form of GLP-1 and the first two amino acids of oxyntomodulin that is also an endogenous ligand.

- We have deleted the sentence for the reasons the reviewer mentions.

Line 329-331 -What data supports that LUXendin555 maintains stable glycaemia?

- We have modified the sentence, as well as included supporting data (lines 310-312):

“Labeling occurred rapidly within 5 min post-injection, produced intense membrane staining confined to the islet where GLP1R is expressed (Fig. 9h), and normoglycemia was not significantly altered over 30 mins (173.0 ± 21.1 versus 215.3 ± 41.4 mg/dl, 0 and 30 min post-injection, respectively; n = 3 mice; non-significant, paired t-test).”

Why were different concentrations of ligands used for the different assays (where single concentrations were assessed)? This should be justified. In addition concentrations of BETP used should also be added to the methods section and figure legends.

- **LUXendin645** was used at 250 nM in MIN6/INS1/AD293 and 50-100 nM in islets. We found that concentrations > 100 nM led to increased background in the latter preparation, probably due to issues washing islets, now stated in the revised manuscript (lines 154-155). For TIRF and STED experiments, **LUXendin645** and **LUXendin651** were titrated to allow single molecule visualization of GLP1R; this is now detailed in the Material and Methods (lines 599-602). For in vivo brain labeling, 100 pmol/g **LUXendin645** was injected subcutaneously, back-calculated from the concentration used in vitro. We have also added **LUXendin555** labeling measurements to the manuscript, showing that max labeling is seen at 600 nM (Figure 9a and b) (lines 290-291). Similarly to **LUXendin645**, we found that a lower concentration (250 nM) of **LUXendin555** gave cleaner staining with less background; however, this extrapolated well to both cells and primary tissue, probably due to physical differences between TMR and Cy5 in cellulose. Again, we have provided better rationale for selecting 250 nM **LUXendin555** in single concentration experiments (lines 291-292). Lastly, we have updated the figure legends to include BETP and **LUXendin** concentrations for all the specific experiments.

Generally the number of individual n numbers for experiments are low. For data that are reported from islets (as number of islets), were these derived from distinct preparations or from one islet isolation? Where cells are reported (ie Figure 2 m n=4-5 images, 57-64 cells), how many individual experiments were performed? This information should be included for each experiment as it speaks to reproducibility. The two-photon in vivo imaging was from 1 experiment on 1 mouse. Is this procedure reproducible?

- To ensure robustness, we have increased the n numbers so that all data are reported from at least two separate islet preparations (i.e. two different days from multiple animals), three independent replicates (for cell line work) or three animals (for in vivo imaging). We also note that five different labs have now received compound and GLP1R KO animals, and report identical results to those presented here. This further reassures us about the robustness of the data and probe. Updated n numbers, specifying the nature of the independent replicate, are presented in the figure legends.

Line 412-418 - Were cAMP assays performed in the presence or absence of IBMX?

- cAMP concentration responses were performed in the presence of IBMX. Measures using Epac2-camps were performed in the absence of IBMX to allow cAMP dynamics to be assessed, rather than accumulation. This has now been stated in the revised manuscript (line 524-525) and we thank the reviewer for pointing this out.

Line 430 – Which cAMP EPAC2 sensor was used or these studies?

- We used Epac2-camps modified with Citrine and Cerulean FRET pairs to minimise pH artefacts associated with YFP variants. The probe was provided by Prof Dermot Cooper (University of Cambridge) and has previously been characterized in ³⁻⁶. This information is included in the revised Material and Methods (lines 539-540).

Line 453 – What concentration of Luxendin was used here?

- MIN6 were treated with 250 nM **LUXendin645**, now stated in the materials and methods.

Figure 1: Peptide sequences do not align to each other (nor does the numbering).

- We apologize for this error, now corrected.

Figures 2 and 6: For the cell surface expression, what are the data normalised to? If this is total GLP-1R expression, what was used for the negative control?

- We have repeated these assays using Ex9 and Ex4 as normalization and positive controls, respectively. Moreover, to differentiate receptor internalization from intracellular fluorophore cleavage (see above), we instead quantified GLP1R using mAb. With this approach, GLP1R surface expression was found to be similar for **LUXendin645**, **LUXendin555** and Ex9 versus Ex4 in islets (Fig. 2g and Fig. 9d), although intracellular punctate staining was still apparent when visualizing the **LUXendin** channel (Supplementary Fig. S3a). We have updated the results as follows (lines 294-299):

“To determine whether the appearance of puncta was due to receptor internalization, or alternatively accumulation of cleaved, charged TMR in organelles, labeling was repeated in islets co-stained with GLP1R monoclonal antibody (Supplementary Fig. S3a). In these experiments, no differences in GLP1R surface expression could be seen between **LUXendin555**, **LUXendin645** and Ex9 (Fig. 9d), suggesting that puncta are unlikely to be internalized receptor.”

Figure 2g and 6d both report on 645, but in fig2 this is around 70 % expression and in 6d 90%. What concentration ligand was used for each? It would be useful to include all ligand concentrations in either the figure or the legend.

- We have now included the ligand concentrations in all figure legends. Following re-analysis to include agonist and antagonist controls, as well as quantification of

monoclonal antibody staining to account for fluorophore cleavage, we don't see any differences in GLP1R surface expression with **LUXendin645** in MIN6 cells versus primary islets (Fig. 2g and Fig. 9d).

Reviewer #3:

The article by Ast et al., "LUXendins reveal endogenous glucagon-like peptide-1 receptor distribution and dynamics", is well-written and the studies presented in the report appear to be of high quality. The key finding is the discovery of far-red fluorescent-containing GLP1R probes that may offer significant advantages over current methods to assess GLP1R expression that use either genetic approaches, or antibody staining/labelled agonist probes. The report clearly presents POC data using pancreatic islets by showing strong membrane labeling for GLP1R, and the authors nicely show loss of the signal in islets from Glp1r knockout mice (as would be expected). A potential advance for the field is the visualization of membrane nano-domains and the ability to track single receptor subpopulations.

- We thank the reviewer for their feedback and for recognising the novelty of the super-resolution studies. We would like to take this opportunity to reiterate the significance of our studies, namely: 1) generation of three bright and highly-specific peptidic labels for visualization of endogenous GLP1R in the nanomolar range using different imaging modes; 2) characterization of the higher organization and single molecule dynamics of a class B GPCR for the first time (using SRRF, STED and SMLM); and 3) application of the probes across multiple tissues, as well as hESC β -like cells, with impact for the wider metabolism, neuroscience and stem cell research communities (see below). As far as we are aware, this high bar has not been achieved elsewhere in the chemical biology field for a GPCR target.

While these tools appear to have great promise, detection of the GLP1R in pancreatic islets is a minimal bar, and the paper would be significantly improved if the technology was applied to characterizing GLP1R expression in the peripheral and central nervous systems and throughout the gastrointestinal tract.

- Primary islets of Langerhans were used throughout, as they provide a good testbed for head-to-head comparison of probe labelling versus the current gold standard monoclonal antibody, which works well in this tissue. The reviewer nevertheless makes a good point about confirming the utility of **LUXendins** in other tissues.

We now provide data showing strong **LUXendin645** labeling in the brain, a major undertaking. Following subcutaneous injection of **LUXendin645**, membrane labeling was specifically localized to neurons in the area postrema (AP)/nucleus tractus solitarii (NTS), arcuate nucleus (ARC)/median eminence (ME), as well as ependymal cells of the choroid plexus (CP). These are all regions known to express GLP1R using reporter and fluorophore-agonist approaches^{7,8}. Notably, **LUXendin645** labeling was restricted to the membrane of GLP1R+ neurons and overlapped with inputs from GLP1-producing neurons, identified using *GLU-YFP* and *GLP1RCre;LSL-GCaMP3* reporter animals. **LUXendin645** allowed super-resolution imaging of GLP1R in the AP, ARC and CP, allowing us to describe the presence of GLP1R nanodomains in the brain for

the first time. Given that monoclonal antibodies do not work reliably in the brain, we predict that the **LUXendins** will make important contributions to our understanding of central GLP1R signaling, as well as interrogation of the neurocircuitry underlying responses to both centrally and peripherally-derived GLP1. The new data are reported in Figure 7, as well as lines 249-268 and lines 375-390 of the revised manuscript.

Moreover, we show that **LUXendin645** labels hESC-derived β -like cells at Day 21 but not Day 0 following initiation of the differentiation protocol. To the best of our knowledge, GLP1R labelling has never before been achieved in this preparation. Moreover, we show that **LUXendin645** allows mature β -like cells to be selected and purified through expression of GLP1R. **LUXendins** therefore open up the possibility to rapidly and inexpensively screen progress of beta cell differentiation in response to various protocols. Again, these findings expand the utility of **LUXendins** into another major research field. The new data are reported in Figure 8, as well as lines 265-274 and lines 269-285 of the revised manuscript.

It is still not completely understood in GLP1R biology how activating there receptor in the periphery and brain impact glucose and energy metabolism. Having the Glp1r knockout mice to use as a negative control would allow exploring the whole animal for the key sites of GLP1 action.

- **LUXendins** are antagonists and as such cannot be used to activate GLP1R signaling in the periphery and brain. In any case, characterization of central and peripheral GLP1R sites involved in glucose homeostasis has already been performed using a range of conditional approaches,⁹⁻¹¹ and is outwith scope of the present manuscript, which focuses on visualizing endogenous GLP1R using chemical biology.

The authors show these probes can be used to assess GLP1R internalization. Therefore, a potential utility of these probes would be to assess GLP1R internalization following treatment with a short-acting versus long-acting GLP1R agonist. Demonstrating differences in GLP1R internalization in animals treated with these types of therapeutic agents could provide insight into what the most desirable profile of a GLP1 medicine would be.

- We cannot perform the suggested experiments, since **LUXendin** requires an unoccupied orthosteric site. Application of a long- or short-acting agonist followed by **LUXendin** labelling would therefore lead to partial agonism/antagonism, poor signal intensity and loss of any receptor internalization. The requested experiments would be best served using recombinant genetic approaches to flexibly tag endogenous receptors with fluorophore. To this end, we have recently generated mice with self-labeling GLP1R and will report these studies elsewhere.

The finding that different fluorophores confer agonistic behavior on the LUX backbone does not indicate better/stabilized incretin therapeutics can be developed. This comment is overstated and should be removed.

- We agree with the reviewer and have deleted the statement in the revised manuscript.

REFERENCES

- 1. Shen, B.-Q. et al. *Nature Biotechnology* **30**, 184-189 (2012).
- 2. Buenaventura, T. et al. *PLOS Biology* **17**(2019).
- 3. Hodson, D.J. et al. *Diabetes* **63**, 3009-3021 (2014).
- 4. Everett, K.L. et al. *PLoS One* **8**, e75942 (2013).
- 5. Willoughby, D. et al. *Journal of Cell Science* **125**, 5850-9 (2012).
- 6. Fine, N.H.F. et al. *Diabetes*, db161356 (2017).
- 7. Richards, P. et al. *Diabetes* **63**, 1224-1233 (2013).
- 8. Secher, A. et al. *Journal of Clinical Investigation* **124**, 4473-88 (2014).
- 9. Varin, E.M. et al. *Cell Reports* **27**, 3371-3384.e3 (2019).
- 10. Holt, M.K. et al. *Diabetes* **68**, 21-33 (2019).
- 11. Sisley, S. et al. *Journal of Clinical Investigation* **124**, 2456-2463 (2014).

REVIEWERS' COMMENTS:

Reviewer #1 (Remarks to the Author):

The authors addressed all my concerns in the revised version. For this reason I support its publication in the current form.

Reviewer #2 (Remarks to the Author):

The authors have performed substantial additional experiments, as well as rewriting sections of the text to address the reviewers comments. The revised version of the manuscript is substantially improved. They have addressed all of my concerns.

Minor comment: Lines 343-347 should be restructured ie to "The following observations support that xxxx...."

Reviewer #3 (Remarks to the Author):

The authors have satisfactorily addressed concerns raised during the initial review of the manuscript. The edits to the text and inclusion of the additional experimental results have improved the paper. No further issues need addressing.

Reviewer #2:

The authors have performed substantial additional experiments, as well as rewriting sections of the text to address the reviewers comments. The revised version of the manuscript is substantially improved. They have addressed all of my concerns.

We are pleased that we have addressed the reviewer's concerns.

Minor comment: Lines 343-347 should be restructured ie to "The following observations support that xxxx...."

The sentence has been modified as follows:

"The following observations support that **LUXendin555** displays antagonist activity in vivo: 1) labeling occurred rapidly within 5 min post-injection, 2) staining was confined to the cell membrane with no apparent internalization (Fig. 9h); and 3) normoglycemia was not significantly altered over 30 mins (173.0 ± 21.1 versus 215.3 ± 41.4 mg/dl, 0 and 30 min post-injection, respectively; n = 3 mice; non-significant, paired t-test)."